

# Increasing flood risk in the Indian Ganga Basin: A perspective from the night-time lights

Ekta Aggarwal[1*], Marleen C. de Ruiter[2], Kartikeya S. Sangwan[1], Rajiv Sinha[3], Sophie Buijs[2], Ranjay Shrestha[4,5], Sanjeev Gupta[1], Alexander C. Whittaker[1]

[1]Department of Earth Science and Engineering, Imperial College London, London, SW72AZ, UK
[2]Institute for Environmental Studies, Vrije Universitet Amsterdam, 1081 HV Amsterdam, The Netherlands
[3]Department of Earth Sciences, Indian Institute of Kanpur, Kanpur, 208016, India
[4]Science Systems and Applications, Inc., Lanham, MD, USA
[5]Terrestrial Information Systems Laboratory, NASA Goddard Space Flight Center, Greenbelt, MD, USA

*Correspondence to*: Ekta Aggarwal (e.aggarwal21@imperial.ac.uk)

* Currently a Research Fellow at the School of Geography and Environmental Science, University of Southampton, SO17 1BJ, UK

**Abstract**

The changing climate, intense rainfall, and geomorphological conditions within the Ganga Basin have led to recurring flooding within the area in the recent past causing severe loss of life and property. The occurrence of such flooding events has increased the need to understand the complex interplay between flood hazards, exposure, vulnerability, and risk. This study delves into flood risk within India's Ganga Basin, focusing on the flood-inducing factors, vulnerability, and exposure through the application of the Analytical Hierarchy Process (AHP) which is a Multi-Criteria Decision Making (MCDM) model. The novelty of the work is using NASA's Black Marble Nighttime Lights as a proxy for human presence and economic activities as an alternative to conventional parameters for flood exposure such as population count, and household density. The study aims to capture the dynamic nature of flood risk, driven by hydro-geomorphic controls, expanding human activities and population growth, and variations in flood resilience. We show that there is a significant increase in flood risk trend in the eastern part of the basin, particularly areas in Bihar, eastern Madhya Pradesh, eastern Uttar Pradesh and the northern part of West Bengal, identifying high flood risk zones at the pixel or cell level. The novelty of the work lies in using night-time lights as a proxy for exposure within the basin, unlike the conventional population data. This study leverages the temporal availability of the data, enabling a real-time distribution of human activities at a large scale and with greater temporal resolution.

The accuracy of the flood risk maps is validated using the historical flood-impacted data from the EM-DAT and GDIS databases, showing a satisfactory model accuracy of approximately 70%. The findings emphasise the role of increasing human exposure and changes in rainfall patterns as the key drivers for increasing flood risk over time. This research has significant implications for flood management, offering insights for developing risk mitigation strategies that transcend administrative boundaries by identifying areas of escalating flood risk.





## 1 Introduction

Floods are one of the most destructive natural hazards causing loss of human life, property, infrastructure, lands and other irreversible damages (IPCC, 2023; Islam et al., 2023; Mishra and Sinha, 2020). As the United Nations Office for Disaster Risk Reduction (UNDRR) reports, the number of flood disasters as reported by the Emergency Events Database (EM-DAT) maintained by the Centre for Research on the Epidemiology of Disasters (CRED) has increased to more than 3200 between 2000-2019 compared to less than 1400 events between 1980-1999, resulting in economic loss worth $651 billion (USD) (Delforge et al., 2023; UNDRR, 2020). The increase in the number of events has been in part attributed to the escalation of extreme hydrological events driven by climate change and anthropogenic activities (Swarnkar et al., 2021; Tripathi and Mohanty, 2024). Moreover, the increase in the severity of flood impact has been complemented by an increase in exposure to flood hazards over time. Floods have varying economic impacts on Low and Middle-Income Countries (LMICs) and High-Income Countries (HICs). In particular, Asian and sub-Saharan African countries have particularly had increased flood exposure between 2000-2015 due to increase in population (Tellman et al., 2021). Large basins in south and southeast Asia (Indus, Ganges-Brahmaputra and Mekong) have some of the largest absolute numbers of people vulnerable to flooding worldwide (17.0–19.9 million, 107.8–134.9 million and 20.2–32.8 million, respectively) and the proportions of their populations exposed to inundation increased by 36%, 26% and 11%, respectively from 2000 to 2015. over [what time?!] (Tellman et al., 2021). The increasingly high economic and human losses associated with flooding make it important to understand the associated risks with extreme events or natural hazards.

Here we analyse the increasing flood risk in the northern part of India, one of the world's most populous regions which is fed by three major rivers – Indus, Ganga and Brahmaputra- and is strongly influenced by the summer monsoon (Immerzeel et al., 2009). These three river basins have a population of over 700 million people, 60% of whom live in the Ganga Basin (Nepal and Shrestha, 2015). The Indian summer monsoon between June to September contributes about 80% of the rainfall in the Ganga Basin (Maurya et al., 2024). Extreme precipitation during the monsoon leads to events such as floods, resulting in significant impacts and extensive devastation to both human lives and livelihoods. Notably, the floods in Uttarakhand (2013), Jammu & Kashmir (2014), Tamil Nadu (2015), Bihar (2016), Gujarat (2017), and Kerala (2018) were among the most devastating events in the past decade, leading to significant socioeconomic disruption, loss of biodiversity, and human lives in India (Mishra and Shah, 2018; Ray et al., 2019; Upadhyay et al., 2020; Swarnkar et al., 2021a, 2021b; Swarnkar and Mujumdar, 2023).

When we study flood risk, the three terms- hazard, exposure and vulnerability are often used interchangeably, leading to different meanings for different users. It is important to understand the role of vulnerability and exposure when assessing flood risk. According to the IPCC (2014) report, flood risk is defined as a product of hazard, exposure and vulnerability based on the common framework adopted by the United Nations, and the definitions of the three risk components are as follows:

1. *"Hazard is defined as the potential occurrence of a natural or human-induced physical event or trend or physical impact that may cause loss of life, injury, or other health impacts, as well as damage and loss to property, infrastructure, livelihoods, service provision, ecosystems, and environmental resources"*

2. *"Exposure is defined as the presence of people, livelihoods, species or ecosystems, environmental functions, services, and resources, infrastructure, or economic, social, or cultural assets in places and settings that could be adversely affected"*,

3. *"Vulnerability is defined as the propensity or predisposition to be adversely affected. Vulnerability encompasses a variety of concepts and elements including sensitivity or susceptibility to harm and lack of capacity to cope and adapt".*



In this study, we constrain all three components separately to assess the long-term flood risk within the basin. This combines
both natural and anthropogenic factors, giving more accurate information about flood risk.
A general overview of different techniques for mapping flood-prone areas is broadly categorised into physical, numerical and
empirical approaches (Liu et al., 2024; Mukhtar et al., 2024; Teng et al., 2017). A physical model approach predicts floods
through a rigorous mathematical treatment based on physical mechanisms, including hydrological parameters, river
geomorphic parameters, and topography by which floods occur (Teng et al., 2017). A numerical modelling approach is
typically based on numerical solutions to solve flow equations in 1-, 2-, and 3-D dimensions and is sometimes referred to as a
hydrodynamic model. An empirical model approach consists of the multi-criteria decision-making method (MCDM) which
uses prior knowledge to predict floods by assuming subjective weights of different factors for potential flood factors, statistical
methods for flood analysis and prediction, and machine learning and AI models.
Over time, there has been an evolution of different methodologies for understanding flood risk management. Integration of
hydrological models, monitoring river gauge data, satellite imagery and remote sensing technologies for forecasting floods,
flood susceptibility mapping and flood inundation mapping namely, have gained popularity. Such methods gather information
on precipitation, river flow, soil moisture, and snowpack, reliable weather forecasts, and advanced meteorological models,
which can be used to predict the amount, intensity, and duration of rainfall in a particular region, aiding in flood predictions
(Islam et al., 2023; Munawar et al., 2022). Analysing flood risk can be a complex and challenging task, mainly due to
limitations in data availability. To date, there are no universal criteria that specify which type of model should be used in which
situation, as each model has pros and cons (Khosravi et al. 2018). However, de Brito and Evers (2016) reviewed almost 128
papers regarding the MCDM model and found that the Analytic Hierarchy Process (AHP) is the most widespread MCDM
method for flood susceptibility modelling. Thus, the AHP method with the integration of remote sensing and GIS provides a
powerful tool for integrating multiple information, preparing flood impact maps, and providing more accurate insights on flood
risk.
The Analytical Hierarchy Process (AHP) developed by Saaty (1980) is one of the widely known approaches for flood
susceptibility mapping (Sinha et al, 2008; Chakraborty and Mukhopadhyay, 2019; Ghosh and Kar, 2018; Grozavu, 2017;
Huang et al., 2011; Mishra and Sinha, 2020). This approach uses pairwise comparisons to assess the extent to which one factor
within the model is more important than the other, thereby producing a weighting for each factor. In the Ganga Basin, previous
research related to flood risk assessment has been conducted in different geographical areas. For instance, Dwivedi (2022)
assessed hydrometeorological risk in the upper reaches of the Ganga Basin using the MaxEnt Machine Learning model. The
study included various parameters derived from a DEM and other sources, albeit without the incorporation of exposure and
vulnerability estimates for the basin. Roy (2021) conducted a flood risk assessment in the active flood-prone regions of the
foothills of the Himalayan range in the Ganga Basin using the Analytical Hierarchy Process (AHP). Their methodology
included various hydrological parameters and flood vulnerability parameters to prepare flood risk assessment in the fan.
Similarly, Mishra and Sinha (2020) and Ghosh and Kar (2018) used AHP to evaluate flood risk in parts of the Ganga basin
based on hydrogeomorphic parameters and socio-economic vulnerability parameters such as population density, household
density, female density, literacy, Land use Land cover (LULC), and road density.
However, these strategies for characterizing regional flood risk have some limitations which are important for us to consider.
They commonly do not consider the three risk components as separate entities, specifically ignoring the vulnerability and
exposure aspects, which are often considered together due to the unavailability of high temporal exposure data. The exposure
data usually used for assessing flood risk in the area is based on the last census data which was back in 2011 (Government of
India, 2011). Secondly, in the case of the Ganga catchment, one of the most populous basins of the world, flood risk assessment
is usually done in parts of the basin and not the basin as a whole. Lastly, the increasing exposure and vulnerability have resulted



in a limited understanding of how risk has changed over time. Thus, in our study, we aim to incorporate factors like
geomorphology, hydrogeomorphic parameters, human exposure and vulnerability of different societies to floods and combine
them using GIS techniques to create a reliable decision-making tool for flood-risk assessment. Since this study is aimed at risk
identification in the Ganga Basin, we take into account easy access to data when determining flood risk factors, that is, to avoid
the actual data acquisition being too complicated or costly. The key novelty of the approach lies in using night-time lights
(NTL) as a proxy for flood exposure within the basin, unlike the population data, and leveraging the temporal availability of
the data. The NTL data can reflect the real-time distribution of human activities at a large scale and with better temporal
frequency, compared to traditional statistics and census data (Fang et al., 2021). It has been investigated as a proxy for human
activities and has been used in various studies for different domains (Andries et al., 2023; Román et al., 2018; Wang et al.,
2018). The data has also increasingly been explored to examine human exposure and presence near rivers, including those
associated with floods (Aggarwal et al., 2024; Ceola et al., 2014; Elshorbagy et al., 2017).
This study aims to conduct a novel flood risk assessment for the Ganga Basin in India at a pixel level, motivated by the
limitations of previous approaches and the emerging opportunities provided by night-time light (NTL) data. By utilising remote
sensing datasets and Geographic Information System (GIS) techniques, and integrating them with the AHP method, this
research presents a detailed evaluation of flood risk. The assessment focuses on key components of flood risk, including hazard,
exposure, and vulnerability, with a specific emphasis on using real-time exposure data from night-time lights.
One of the key contributions of this study is the development of annual flood risk maps for the Ganga Basin from 2013 to
2023. These maps provide a valuable tool for identifying areas where flood risk has been increasing over time, down to the
pixel or cell level. This spatial and temporal analysis offers a clearer picture of evolving flood risks in the basin, revealing
trends and highlighting regions that may require immediate attention. The approach developed here is designed to assist a
range of stakeholders, from local communities to policymakers, in creating more effective flood risk management strategies.
The annual flood risk maps, in particular, are anticipated to inform these stakeholders by identifying areas with rising flood
threats and guiding efforts to reduce the impacts of flooding. This, in turn, could help minimize loss of life, economic damage,
and disruptions to society, making the findings of this research crucial for future flood management planning in the region.



**2   Study Area**

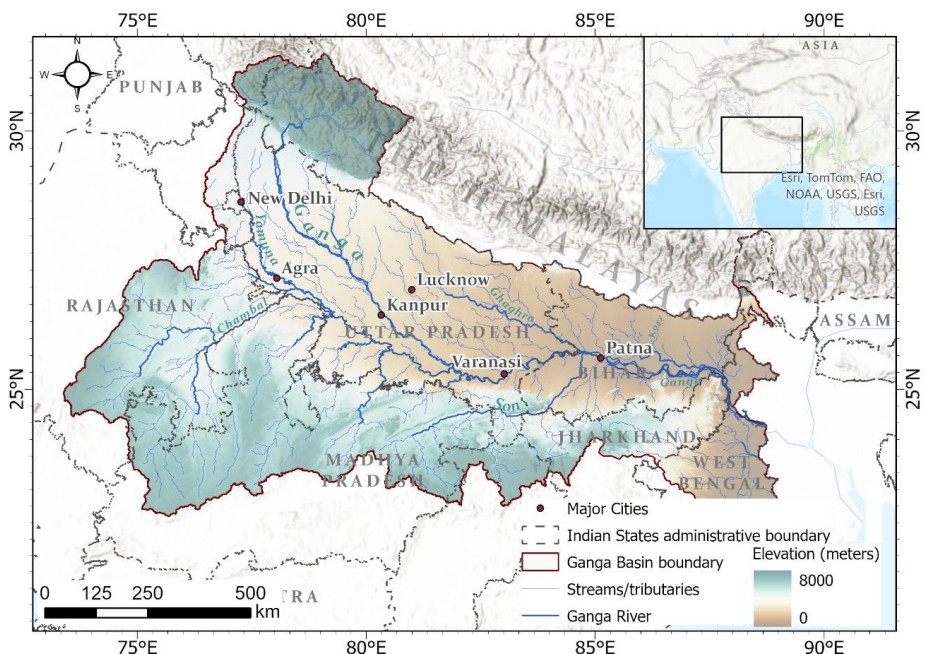


**Figure 1: ESRI world imagery showing the Ganga Basin (dark red boundary), Ganga River and its major tributaries in dark blue**
**in India. The other streams are light blue. The inset in the top-right of the figure shows the location of the study area within Asia.**
The Ganga River basin is one of the biggest in the world, spread over India, Bangladesh, Nepal and Tibet. In our study, we
focus on the Indian part of the basin which is the largest river system in India (Figure 1) covering an area of ca. 860,000 km²
draining into 11 states of the country - Uttarakhand, Uttar Pradesh, Haryana, Himachal Pradesh, Delhi, Bihar, Jharkhand,
Rajasthan, Madhya Pradesh, Chhattisgarh and West Bengal (River Basin Atlas of India, 2012). The basin covers > 26% of the
total geographic area of India (Tripathi and Mohanty, 2024) and extends between the latitude of 21°6'N to 31°21'N, and the
longitude of 73°2'E to 89°5'E. The Himalayas bound the basin to the north and the primary river of the basin is the Ganga (or
Ganges), which originates from the Gangotri Glacier in the Himalayas at an elevation of nearly 7010 m and traverses a length
of > 2500 km before it flows into the Bay of Bengal (Figure 1). Along its way, the Ganga is joined by several tributaries to
form the most fertile alluvial plain in Northern India (River Basin Atlas of India, 2012). A few of the important tributaries
include the Yamuna, Chambal, Kosi and Ghaghara. All the rivers in the basin are perennial and carry large runoff due to heavy
rainfall, making it one of the most flood-prone regions in the world (Swarnkar et al., 2021b). The basin receives rainfall during
the southwest monsoon between June to October. In the majority of the basin, the average rainfall is 70-80% of the total annual
rainfall over three months from July to September. The annual average rainfall in the basin varies between 400 - 2000 mm
with approximately 179 and 152 rainy days in the upper and lower basins (River Basin Atlas of India, 2012). In the eastern
part of the basin such as areas of West Bengal and Bihar, the monsoon is longer starting from early June to early October. The
effect of the monsoon weakens from east to west.
**3   Data & Methods**
Flood risk assessment considers not only the hydrological extremes but includes complex processes characterised by a large
number of physical hazards, vulnerability and exposure parameters. Consequently, both temporal and spatial resolutions are
essential for comprehensive analysis. To develop variable risk maps for the Ganga Basin, we employed multiple datasets and





adhered to a rigorous methodology to identify areas of prolonged risk. Figure 2 illustrates the workflow of the research. The
first step was acquiring various data which are listed in Table 1. The next step was processing and calculating normalised
relative weights for each factor to compute flood hazard, exposure and vulnerability respectively. This was followed by
computing the flood risk, validating the data model and calculating the flood risk trend. Below we outline these data-sets and
processing steps in further detail, which are shown diagrammatically in Figure 2.

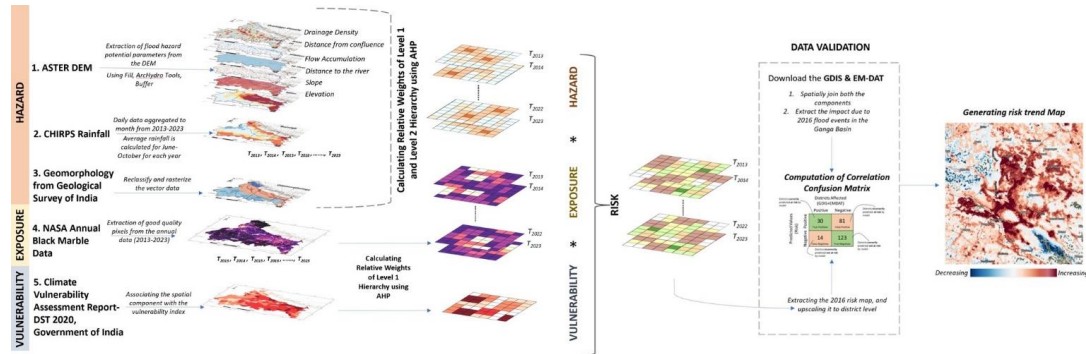


**Figure 2: Flow chart illustrating the methodology for this work.**
**3.1    Datasets**

**Table 1: List of datasets used in the study and their sources**

|   | Dataset | Source |
|---|---------|--------|
| 1 | Digital Elevation Model (DEM) | ASTER-GDEM (https://appeears.earthdatacloud.nasa.gov/) |
| 2 | Rainfall | Climate Hazards Group InfraRed Precipitation with Station data (CHIRPS) ((https://developers.google.com/earth-engine/datasets/catalog/UCSB-CHG_CHIRPS_DAILY#description) |
| 3 | Geomorphology | Geological Survey of India (https://bhukosh.gsi.gov.in/Bhukosh/Public) |
| 4 | Exposure | NASA Black Marble nighttime lights annual product suite (VNP46) (https://ladsweb.modaps.eosdis.nasa.gov/search/order/2/VNP46A4--5000 ) |
| 5 | Vulnerability | Climate Vulnerability Assessment Report- DST 2020, Government of India (https://dst.gov.in/national-climate-vulnerability-assessment-identifies-eight-eastern-states-highly-vulnerable) |
| 6 | Historic Disaster Impact Database | EM-DAT (https://public.emdat.be/); GDIS (https://sedac.ciesin.columbia.edu/data/set/pend-gdis-1960-2018) |

1.  **Digital Elevation Model (DEM):** We used the free open source DEM i.e., ASTER-GDEM (Advanced Spaceborne

Thermal Emission and Reflection Radiometer Global Digital Elevation Model) Version 2. The spatial resolution of

the data is 1 arc-second, approx. 30m spatial resolution (NASA/METI/AIST/Japan Spacesystems and U.S./Japan

ASTER Science Team, 2019). The DEM is further used for extraction of elevation, slope, drainage density, distance

from the confluence, flow accumulation and distance to the river which are further explained in section 3.3.

2.  **CHIRPS Rainfall Data:** Climate Hazards Group InfraRed Precipitation with Station data (CHIRPS) is a 35+ year

quasi-global rainfall data set. The data spans 50°S-50°N (and all longitudes) and ranges from 1981 to the present.

CHIRPS incorporates the climatology, CHPclim, 0.05° resolution satellite imagery, and in-situ station data to create

gridded rainfall time series (Funk et al., 2015). The latest version 2.0 (V2.0) of this precipitation data set is available



| | |
|---|---|
| 180 | for 1981–2017 at 0.05° spatial resolution and at daily, pentad, and monthly time scales. In our study, we used the |
| 181 | daily data from 2013-2023 accessed using Google Earth Engine. |
| 182 | 3. **Geomorphology:** To incorporate the geomorphology in assessing the potential flood hazard, we used the |
| 183 | geomorphology information provided by the Geological Survey of India, which can be accessed online on the |
| 184 | Bhukosh server (https://bhukosh.gsi.gov.in/Bhukosh/Public). We use the Geomorphology 250k vector data which |
| 185 | provides spatial information on the floodplains, alluvial plains, hills and valleys, plateaus and pediment-peneplain |
| 186 | complex. |
| 187 | 4. **NASA Black Marble Annual Data Product:** NASA Black Marble nighttime lights product suite (VNP46) produced |
| 188 | at NASA's Land Science Investigator-led Processing Systems (Land SIPS) is a product of the Visible Infrared |
| 189 | Imaging Radiometer Suite (VIIRS) Day/Night Bands (DNB) available from January 2012 at daily/monthly/yearly |
| 190 | frequency (Román et al., 2018). The data captures human signal-artificial signals at night, providing an understanding |
| 191 | between the human systems and the environment. The product suite, available globally on daily, monthly, and annual |
| 192 | composite scales, corrects extraneous sources of noise in nighttime light (NTL) radiance signals (Román et al., 2018). |
| 193 | This study uses the VNP46A4 yearly product. The data has a spatial resolution of 15 arc seconds (approx. 500m) and |
| 194 | is provided in the Hierarchical Data Format Earth Observing System (HDF-EOS) format (Román et al., 2018). The |
| 195 | product has 28 layers. In our analysis, we use the 'AllAngle_Composite_Snow_ Free' and |
| 196 | 'AllAngle_Composite_Snow_ Free_Quality' bands. The NTL radiance is measured in nWatts/cm$^2$sr units having a |
| 197 | valid range of 0-65534, and the fill value for no data is 65535. The quality flag layer gives information about the |
| 198 | quality of the NTL pixel retrieved at the time of acquisition. The quality pixel values are unitless and range from 00- |
| 199 | 02, where 255 is the fill value for no data. 00 are values of high quality, 01 denotes pixels with bad quality and 02 are |
| 200 | the gap-filled quality pixels (Román et al., 2018). |
| 201 | 5. **Climate Vulnerability Assessment Report- DST 2020:** For assessing the vulnerability, we used the report |
| 202 | developed by the Department of Science and Technology, Government of India (https://dst.gov.in/national-climate- |
| 203 | vulnerability-assessment-identifies-eight-eastern-states-highly-vulnerable, last access: 3$^{rd}$ March 2024). The report |
| 204 | presents the vulnerability index for each district and state in the country. The vulnerability index is calculated using |
| 205 | 14 indicators based on biophysical, socio-economic, and institution and infrastructure-related vulnerability indicators, |
| 206 | each with an equal weight. The indicators capture both sensitivity and adaptive capacity (Climate Vulnerability |
| 207 | Assessment for Adaptation Planning in India Using a Common Framework, 2020). The vulnerability data from the |
| 208 | report has been previously used for assessing basin risks in India (Vegad et al., 2024). |
| 209 | 6. **GDIS and EM-DAT:** EM-DAT is the International Emergency Event Database maintained by the Centre for |
| 210 | Research on the Epidemiology of Disasters (CRED), containing data on the occurrence and impacts of over 26,000 |
| 211 | mass disasters worldwide from 1900 to the present day (Delforge et al., 2023). The database is compiled from various |
| 212 | sources, including UN agencies, non-governmental organizations, reinsurance companies, research institutes, and |
| 213 | press agencies. It documents disasters as a single group, which contains five subgroups – geophysical, meteorological, |
| 214 | hydrological, climatological, and biological – each of which further contains one or more types of specific natural |
| 215 | hazards, and their physical impacts or risks – fatality, injured, affected, and damage are described (Delforge et al., |
| 216 | 2023). Geocoded Disasters (GDIS) is an open-source extension to the Emergency Events Database (EM-DAT) that |
| 217 | allows researchers, for the first time, to explore and make use of subnational, geocoded data on major disasters |
| 218 | triggered by natural hazards (Rosvold and Buhaug, 2021). The GDIS dataset provides spatial geometry in the form |
| 219 | of GIS polygons and centroid latitude and longitude coordinates for each administrative entity listed as a disaster |
| 220 | location in the EM-DAT database. In total, GDIS contains spatial information on 39,953 locations for 9,924 unique |
| 221 | disasters occurring worldwide between 1960 and 2018. The dataset facilitates connecting the EM-DAT database to |



other geographic data sources on the sub-national level to enable rigorous empirical analyses of disaster determinants
and impacts.
**3.2    Analytical Hierarchy Process (AHP) Framework**
When different thematic layers are combined on a GIS platform, a Multi-Criteria Decision Analysis or Multi-Criteria Decision
Making can be used to make an evaluation based on several criteria and disagreeing evaluations (Sinha et al., 2008; Huang et
al., 2011). One of the widely used MCDMs is the Analytical Hierarchy Process (AHP) developed by Saaty (1980) which
incorporates both practical and subjective knowledge. The method involves decomposing the problem into a hierarchy and
calculating the relative weight percentage in a GIS environment. The AHP model consists of four stages, namely (i) parameter
hierarchy construction, (ii) pairwise comparison matrix, (iii) weight normalisation, and (iv) consistency check (Ghosh and Kar,
2018; Mishra and Sinha, 2020; Roy et al., 2021). The first step is the selection of parameters for flood hazard, exposure and
vulnerability assessment respectively. This is usually based on the available literature and discussion among researchers. In
this study, we include eight parameters for assessing flood hazard namely- elevation, slope, flow accumulation, drainage
density, distance to the river, distance from the confluence, rainfall and geomorphology as outlined in section 3, and we further
return to the significance of these factors in the results below. For studying the exposure, we chose to use the radiance values
from the night-time light data signifying activities and district vulnerability index from the Climate Vulnerability Assessment
Report- DST 2020, Government of India for calculating flood vulnerability in the catchment. The second step is developing
the pairwise comparison matrix using the scale developed by Saaty 1980 at each decision level (Table 2). This scale utilises a
basic sequence of absolute numbers from 1 to 9 for each pair to represent the individual preferences in the upper half of the
matrix. In the lower half of the matrix, the pairing is assigned a rating equal to the reciprocal of the value of the corresponding
pair in the upper matrix based on the decision maker's subjectivity, experience, and knowledge intuitively and naturally (Saaty,
2008). Following the construction of a pairwise comparison matrix, weights are calculated and a consistency check is done.

**Table 2: Relative importance scale (1–9) developed by Saaty (1980,1990)**

| Intensity of Importance | 1 | 2 | 3 | 4 | 5 | 6 | 7 | 8 | 9 |
|---|---|---|---|---|---|---|---|---|---|
| Definition | Equal | Weak | Moderate | Moderate Plus | Strong | Strong Plus | Very Strong | Very very strong | Extreme |


**Table 3: Random consistency index (RCI) based on by Saaty (1980,1990)**

| No. of parameters selected | 1 | 2 | 3 | 4 | 5 | 6 | 7 | 8 | 9 | 10 |
|---|---|---|---|---|---|---|---|---|---|---|
| RCI Value | 0 | 0 | 0.58 | 0.90 | 1.12 | 1.24 | 1.32 | 1.41 | 1.45 | 1.49 |


The following is the series of steps for the computation of the pairwise matrix:
*Step1*: Giving pairwise priorities ($X_{11}$, $X_{12}$, …$X_{nn}$) based on Table 2 for each parameter ($P_1$, $P_2$,…$P_n$) and summing the priorities
for each column.

|  | $P_1$ | $P_2$ | $P_n$ |
|---|---|---|---|
| $P_1$ | $X_{11}$ | $X_{12}$ | $X_{1n}$ |



| P₂ | X₂₁ | X₂₂ | X₂ₙ |
|---|---|---|---|
| **Pₙ** | Xₙ₁ | Xₙ₂ | Xₙₙ |
| | **S₁** | **S₂** | **Sₙ** |

Thus, the Sum of priorities (Sₙ) for the nᵗʰ Parameter (Pₙ) :

$$S_n = X_{1n} + X_{2n} + \ldots + X_{nn} \quad\quad \ldots\ldots\ldots\ldots\ldots\ldots\ldots\ldots\ldots\ldots \text{ (1)}$$

*Step 2*: Normalise each value by dividing each cell ($X_{nn}$) in Step 1 with the column total ($S_n$) computed from equation 1.

| | **P₁** | **P₂** | **Pₙ** |
|---|---|---|---|
| **P₁** | X₁₁/S₁ | X₁₂/S₂ | X₁ₙ/Sₙ |
| **P₂** | X₂₁/S₁ | X₂₂/S₂ | X₂ₙ/Sₙ |
| **Pₙ** | Xₙ₁/S₁ | Xₙ₂/S₂ | Xₙₙ/Sₙ |

*Step 3*: Computing the relative weightage for each element ($RW_n$) by addition of row-wise priorities (Equation 2).

| | **P₁** | **P₂** | **Pₙ** | |
|---|---|---|---|---|
| **P₁** | X₁₁/S₁ | X₁₂/S₂ | X₁ₙ/Sₙ | **RW₁** |
| **P₂** | X₂₁/S₁ | X₂₂/S₂ | X₂ₙ/Sₙ | **RW₂** |
| **Pₙ** | Xₙ₁/S₁ | Xₙ₂/S₂ | Xₙₙ/Sₙ | **RWₙ** |

$$RW_n = (X_{n1}/S_1) + (X_{n2}/S_2) + \ldots + (X_{nn}/S_n) \quad \ldots\ldots\ldots\ldots\ldots\ldots\ldots\ldots\ldots \text{ (2)}$$

*Step 4*: Calculating the Consistency Ratio (CR) to validate the AHP judgement

$$CR = CI/RI \quad\quad \ldots\ldots\ldots\ldots\ldots\ldots\ldots\ldots\ldots\ldots \text{ (3)}$$

$$CI = (\lambda_{max} - n)/(n-1) \quad\quad \ldots\ldots\ldots\ldots\ldots\ldots\ldots\ldots\ldots \text{ (4)}$$

where CI is the Consistency Index, RI is the Random Consistency Index based on Table 3, $\lambda_{max}$ is the principle Eigenvalue and n indicates the total number of parameters selected for the study.

Thus, to compute the consistency ratio, we first derive the weighted sum ($WS_n$) for each row. To compute it, we multiply the relative weightage (equation 2) for each element computed in step 3 and the pairwise priorities derived in step 1.

| | **P₁** | **P₂** | **Pₙ** | |
|---|---|---|---|---|
| **P₁** | X₁₁*RW₁ | X₁₂*RW₂ | X₁ₙ*RWₙ | **WS₁** |
| **P₂** | X₂₁*RW₁ | X₂₂*RW₂ | X₂ₙ*RWₙ | **WS₂** |
| **Pₙ** | Xₙ₁*RW₁ | Xₙ₂*RW₂ | Xₙₙ*RWₙ | **WSₙ** |

Second, $\lambda_{max}$ is calculated based on the following computation:

$$\lambda_{max} = [(WS_1*W_1) + (WS_2*W_2) + \ldots + (WS_n*W_n)]/n \quad \ldots\ldots\ldots\ldots\ldots\ldots\ldots \text{ (5)}$$



Lastly, the Consistency Index is computed based on equation 4. Lastly, the Consistency Ratio (CR) for each AHP judgment is
calculated based on Equation 3.

### 3.3    Defining factors that contribute to flood hazard

AHP requires the identification of factors that contribute to the flood hazard. We defined factors for mapping flood-prone areas
or flood hazards based on literature analysis and the characteristics of the study area. Previously many researchers have
undertaken flood hazard mapping using a combination of geomorphic morphometrics parameters, hydrological analysis, slope,
aspect, type, curvature, precipitation index, rainfall, land use land cover (LULC), distance to roads, topographic roughness
index (TRI), topographic wetness index (TWI), steam power index, sediment transport Index (STI) amongst others (Adnan et
al., 2019; Kanani-Sadat et al., 2019; Towfiqul Islam et al., 2021). In our study, the key factors we used to map the flood hazard
in the basin are – elevation, slope, flow accumulation, drainage density, distance to the river, distance from the confluence,
rainfall and geomorphic setting, and are shown in Figure 3. The figure shows the potential occurrence of the floods with respect
to each parameter. The reasoning of the selection of the factors are further discussed in the section 4.1.
The first dataset we used to derive potential flood hazard was the ASTER 30m DEM. With a sink-filled DEM as the input, the
slope was computed using the Slope function in ArcGIS Pro. Further, flow accumulation, drainage density and stream network
were calculated using the ArcHydro Tools in the ArcGIS Pro. Our analysis focuses on the main river (Ganga) and its major
tributaries, considering a minimum stream order of 5 for the analysis. To evaluate the proximity to the river, we created seven
buffer zones at intervals of 0-500m, 500-1000m, 1000-1500m, 1500-2000m, 2000-2500m, 2500-3000m, and 3000-3500m
using the buffer tool in the Geoprocessing toolbox in ArcGIS Pro. Similarly, the buffer zones for the confluence points were
constructed at intervals of 0-500m, 500-1000m, 1000-1500m, 1500-2000m, and 2000-3000m respectively.
For rainfall analysis, we used the daily data CHIRPS raster data to prepare the monthly average composite. Furthermore,
monthly data from 2013 to 2023 was analysed focussing on the monsoon months from June to October due to a higher
frequency of flooding recorded during this period. This contrasts with other flood risk studies which only consider average
annual rainfall for risk analysis. Thus, we used this data to generate annual flood risk maps for each subsequent year. The
exported monthly composite was clipped to the basin boundary.
The geomorphic setting of the basin was downloaded from the Bhukosh server (https://bhukosh.gsi.gov.in/Bhukosh/Public).
The data was downloaded in a vector format and contains a breakdown of basin elements including fluvial, structural,
denudational and aeolian classes, as well as landforms such as hills, valleys, plateaus, and alluvial plains. The actual categories
in the data are – FluOri- Active floodplain (i.e., Fluvial origin active floodplain), Fluori- Older floodplain, Fluori- Younger
alluvial plain, Fluori - Older alluvial plain, Denori- Moderately dissected lower plateau (i.e., denudational origin), StrOri-
Highly dissected hills and valleys (i.e., structural origin), Strori- Moderately dissected hills and valleys, Strori- Low dissected
hills and valleys and StrOri- Pediment-Peneplain Complex respectively. The vector data categories were reclassified based on
the geomorphology: active floodplains, older floodplains, younger alluvial plains, older alluvial plains, hills and valleys,
plateaus and pediment-peneplain complexes. The data was also clipped to the basin boundary. Figure 3 shows the potential
hazard parameters data that are used to prepare for the potential flood hazard. All the hazard parameters raster layers were
resampled to the 500 m spatial resolution for further integration with other satellite datasets.
Supplementary Table 1 shows the relative weight for each parameter calculated based on the above AHP Procedure from Step
1 to Step 3 in section 3.2. Further, for each parameter, a level 2 hierarchy was structured. Supplementary Table 2 to
Supplementary Table 9 shows the pairwise construction matrix and relative weights for sub-factors for elevation, slope,
rainfall, distance to the river, distance to the confluence points, drainage density, flow accumulation, and geomorphology
respectively. The level 1 and level 2 hierarchy for calculating FHI was constructed based on a literature review and discussions



on the understanding of the flood-prone areas, which is in detail discussed in the section 4.1. Following the construction of a
pairwise matrix and the calculation of relative weights at level 1 and level 2 hierarchy, based on Step 4 of section 3.2, the $\lambda_{max}$
principle Eigenvalue, the consistency index (CI) and the consistency ratio (CR) were calculated for the hazard factors
(Supplementary Table 1 to Supplementary Table 9). This was followed by normalising the relative weights by multiplying the
relative weights (RW) of level 1 and level 2 hierarchy.
Finally, the flood Hazard Index (FHI) at each pixel location *(i)* is calculated based on the following equation:
$FHI_i = (RW_E * E_i) + (RW_S * S_i) + (RW_R * R_i) + (RW_{DR} * DR_i) + (RW_{DC} * DC_i) + (RW_{DD} * DD_i) + (RW_{FA} * FA_i) + (RW_G * G_i)$
.................................................................*(6)*
, where FHI is Flood Hazard Index at pixel *i*, RW are the relative weights calculated from Supplementary Table 1 for level 1
parameters, $E_i$ is the elevation at pixel *i*, $S_i$ is the slope at pixel *i*, $R_i$ is the rainfall at pixel *i*, $DR_i$ is the distance to the river at
pixel *i*, $DC_i$ is the distance to the confluence at pixel *i*, $DD_i$ is the drainage density at pixel *i*, $FA_i$ is flow accumulation at pixel
*i* and $G_i$ is geomorphology at pixel *i*.





**Figure 3: The figure illustrates eight potential flood hazard parameters, which are (a) elevation, (b) slope, (c) flow accumulation, (d) drainage density, (e) distance to the river, (f) distance from the confluence, (g) rainfall and (h) geomorphology. The colour scheme of blue to red is adopted to indicate the potential occurrence of floods. The blue area shows a low potential for the occurrence of floods compared to the red region which shows a high potential for flood hazard.**



### 3.4 Mapping Yearly Exposure to flood hazard

For mapping the yearly exposure from 2013 to 2023, we used the annual dataset from the NASA Black Marble data. The night-time lights provide a significant proxy for human presence as described by previous researchers (Aggarwal et al., 2024; Andries et al., 2023; Fang et al., 2021; Román et al., 2018). The NTL and quality layers were extracted, converted to tiff format, and mosaicked using Python script. The good-quality pixels of the NTL were extracted and clipped to the basin boundary. Figure 4 shows the distribution of the night-time lights averaged for the period 2013 to 2023. Further, the intensity for the annual data was classified into 5 categories based on geometric classification, i.e., 0-10, 10-17, 17-34, 34-293 and >=293 $nW/cm^2sr^1$. The Supplementary Table 10 shows the pairwise matrix table for the sub-factor and the relative weights that were computed. Based on Step 4 of the section 3.2, the $\lambda_{max}$ principle Eigenvalue, the consistency index (CI) and the consistency ratio (CR) were calculated to check the decision of the factors. The annual exposure rasters are reclassified to the calculated relative weights, which would represent the Flood Exposure Index (FEI) at each pixel.

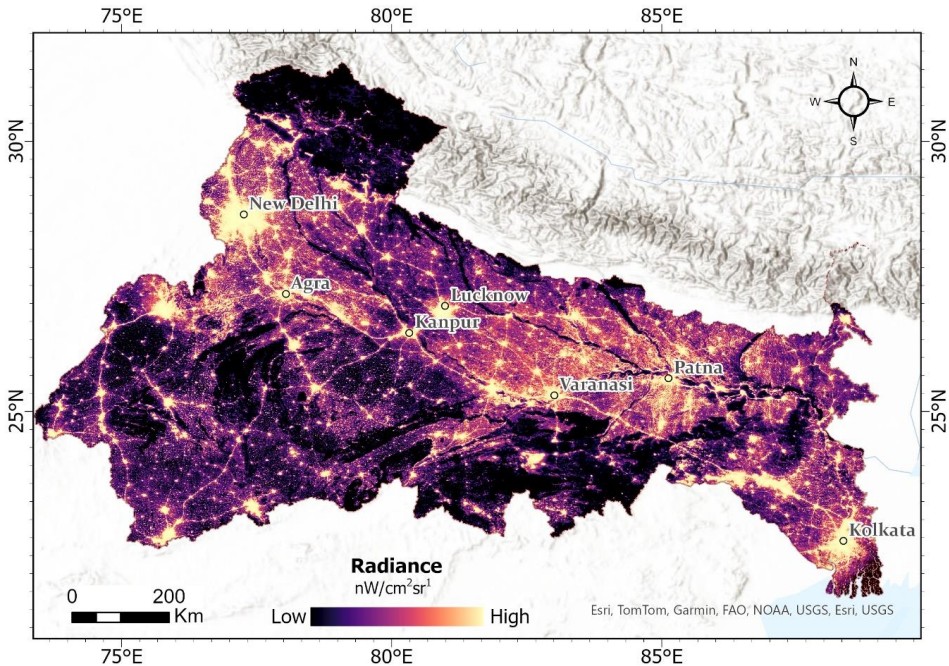

**Figure 4: Distribution of average night-time lights indicating exposure from 2013-2023. The dark areas show the absence of night-time lights and the bright areas are where high human activity-related artificial lights are captured.**

### 3.5 Extracting Vulnerability to flood hazard

The vulnerability index used from the DST vulnerability assessment report is used to identify flood risk in the basin. The report provides the vulnerability index in tabular format at the district level. We incorporated these index values as attributes into the district shapefile for spatial analysis, which was then clipped to the Ganga basin boundary. In Figure 5 we observe that the vulnerability index values from the Climate Vulnerability Assessment Report- DST 2020, Government of India, range from 0.35 to 0.72 within the basin. The index values are divided into 5 categories- 0.35-0.425, 0.425-0.5, 0.5-0.575, 0.575-0.65 and 0.65-0.73 based on the index classification in the report. New Delhi territory is not included in the vulnerability assessment report because of high human presence and activities. For further analysis for risk assessment, the spatial vulnerability vector data was converted into a raster format at a spatial resolution of 500m. Supplementary Table 11 shows the pairwise matrix



table for the sub-factor and the computed relative weights. Based on Step 4 of section 3.2, the $\lambda_{max}$ principle Eigenvalue, the
consistency index (CI) and the consistency ratio (CR) were calculated to check the decision of the factors. Lastly, the
vulnerability index raster is reclassified to the calculated relative weights, which would represent the Flood Vulnerability Index
(FVI) at each pixel.

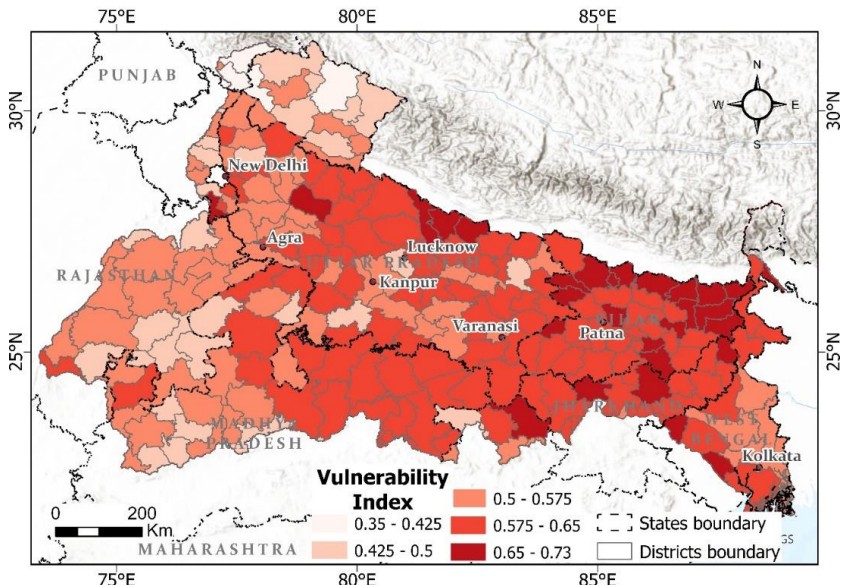


**Figure 5: Vulnerability index of the districts in the Ganga Basin from the Climate Vulnerability Assessment Report- DST 2020,**
**Government of India.**

### 3.6    Flood Risk Index (FRI)

To compute the flood risk at the pixel or cell level for each year *(t)*, we calculated the Flood Risk Index (FRI) as the product
of the Flood Hazard Index (FHI), Flood Exposure Index (FEI) and Flood Vulnerability Index (FHI), as shown below:
$$FRI_t = FHI_t * FEI_t * FVI \qquad \ldots\ldots\ldots\ldots\ldots\ldots\ldots\ldots\ldots\ldots\ldots\ldots\ldots\ldots(7)$$

The FHI, FEI, and FVI represent the normalized relative weights of each parameter at each pixel, obtained through AHP
classification as outlined in section 3.2. The FHI was determined using a two-level hierarchical structure, with normalized
weights derived from Equation 6. Similarly, a Level 1 hierarchy was employed to compute the relative weights for FEI and
FVI at each pixel, as discussed in sections 4.3 and 4.4 respectively.
The flood risk index for each year was then obtained by multiplying the corresponding FHI, FEI, and FVI rasters for each pixel
from 2013 to 2023. These indices were classified into five risk levels. Figure 2 illustrates the process of multiplying these
components. After generating the annual risk maps, we used the 10-year dataset to calculate the flood risk trend for each pixel.
The trend raster was computed using linear regression, implemented through a Python script. This enabled us to calculate a
percentage point change in FRI per year, enabling us to identify areas of significantly increasing flood risk.




**3.7    Data Validation**

To validate our risk maps we compared them with an impact-based dataset using the ROC-AUC (Receiving Operating Characteristic-Area Under Curve) methodology. The ROC is the probability curve and the AUC is the measure of the separability of different classes (Nam and D'Agostino, 2002). This popular method is used for quantitative validation (Das, 2020; Lin et al., 2019; Mukhtar et al., 2024; Roy et al., 2021; Saha and Agrawal, 2020). The ROC curve evaluates the effectiveness of a flood risk model by graphing the true positive rate against the false positive rate. The AUC value represents the model's capacity to distinguish between flood-risk districts and non-flood-risk districts, with higher values reflecting greater accuracy. This validation metric serves as evidence of the flood risk map's robustness, offering a quantitative assessment of its reliability in predicting and classifying flood risk within the studied area.

We used the AHP-derived flood risk map to compute the absolute risk index values for each district within the basin and the spatial impact flood inventory map at the district level. In this method, a confusion matrix was computed stating the true positive, false positive, true negative, and false negative at different risk value thresholds. The definitions of the four values are described below:

1. True positives (TP): captures the number of districts at *'risk'* that are also predicted as *'impacted'* by floods in the EM-DAT/GDIS derived data.

2. False positives (FP): captures the number of districts *'not at risk'* that are predicted as *'impacted'* by floods in the EM-DAT/GDIS derived data.

3. True negatives (TN): captures the number of districts *'not at risk'* that are predicted as *'not impacted'* by floods in the EM-DAT/GDIS derived data.

4. False negatives (FN): captures the number of districts at *'risk'* that are also predicted as *'not impacted'* by floods in the EM-DAT/GDIS derived data.

For constructing the spatial impact flood inventory map, we utilised both GDIS and EM-DAT for validation. First, we downloaded the GDIS data from https://sedac.ciesin.columbia.edu/data/set/pend-gdis-1960-2018 (accessed on 25th April 2024). The geodatabase data (.gdb) includes both spatial and attribute table components. The data was clipped to the Ganga basin boundary in ArcGIS Pro giving spatial information at the administrative level for all disaster events, with each disaster assigned a unique disaster number. Next, EM-DAT data was downloaded from the https://public.emdat.be/portal (accessed on 5th March 2024), which is a global repository of all disaster events. The downloaded data included flooding events in India from 2013 to 2023 and contained information on the total number of affected people, linked to the unique disaster numbers also found in the GDIS data.

The two datasets (i.e., EM-DAT csv data & the GDIS shapefile) were merged based on the unique disaster number using a Python script resulting in an updated GDIS shapefile that included the total number of people affected in its attribute table. Additional tools, such as Multipart to Singlepart and Pairwise Dissolve, were used for data processing and refinement. A new column, Area (km$^2$) was added representing the spatial polygon area. This processed data repository consisted of all the flooding events from 2013 to 2018, providing information on the affected districts in a vector format, aligning with the common timeline of both data. For a case study, we chose the 2016 flooding event and created a new file giving information on the affected districts, polygon areas and the total number of people affected by a particular disaster. We chose the 2016 specific case study because of two primary reasons. First, the common availability of years between the GDIS and the risk map generated is between 2013 and 2018. Secondly, as per the Global Flood Database, the flood event triggered due to heavy rains on 25th July 2016 had the highest exposed population of 18,456,496 million between 2013 to 2018 in the Ganga Basin (Tellman et al., 2021). Thus, justifying our choice of the 2016 flood risk as a case study for validating the data. The major limitation of linking the spatial data with the affected population was that the number of people listed for each administrative level



represented the total number of people affected in a particular disaster number. Thus, to address this issue, we normalised the number of people affected for each district based on the district area and used the 2016 event, consisting of 46 entries, to validate the risk maps.

Next, we converted the 2016 risk map to a vector format at the district level for two primary reasons: to reconcile the format discrepancy between the raster risk map and the vector impact data; and secondly, because the impact maps operate at a district-level spatial scale. For comparison, we calculated the zonal risk index for each district in the Ganga Basin followed by the vectorisation of the data. This processing of the risk maps was done using ArcGIS Model Builder on ArcGIS Pro (Figure 6). Supplementary Table 10 lists out the risk index and people affected by the floods in 2016 for each district in the Ganga Basin.

Lastly, for the construction of the confusion matrix and deriving the ROC-AUC curve, we computed the True Positives (TP), False Positives (FP), True Negatives (TN) and False Negatives (FN) districts. Figure 7 shows the schematic illustration of the four components computed between the districts predicted at risk by AHP and districts affected by the GDIS and EMDAT database. We performed a spatial join between the 2016 district risk map and the EM-DAT/GDIS-derived 2016 flood impact district map and calculated the normalised number of people affected based on the area of the respective district. Figure 7 schematically illustrates the four components of the confusion matrix and their definitions.

Finally, to assess the effectiveness of our risk model using the ROC-AUC curve, the true positive rate (TPR) or sensitivity and the false positive rate (FPR) or 1-specificity are calculated, which are defined as –

$$\textit{True Positive Rate (TPR) or Sensitivity} = TP/(TP+FN) \qquad \text{............. (8)}$$

$$\textit{False Positive Rate (FPR) or 1-Specificity} = FP/(FP+TN) \qquad \text{............(9)}$$

The ROC-AUC curve assesses the performance of a flood risk model by plotting the true positive rate along the y-axis against the false positive rate on the x-axis. An AUC value signifies the model's ability to discriminate between predicted risk districts and actual flood impact districts, with higher values indicating better accuracy. The quantitative-qualitative relationship between AUC and prediction accuracy, which ranges from 0 to 1, is as follows: Excellent (0.9–1), very good (0.8–0.9), good (0.7–0.8), moderate (0.6–0.7) and weak (0.5–0.6) (Yesilnacar and Topal, 2005).

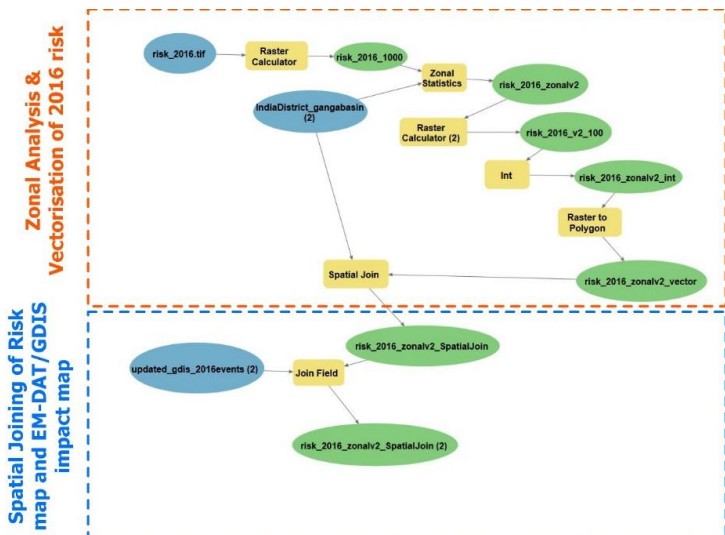

**Figure 6: ArcGIS Model builder showing the workflow for the processing of the 2016 risk map.**



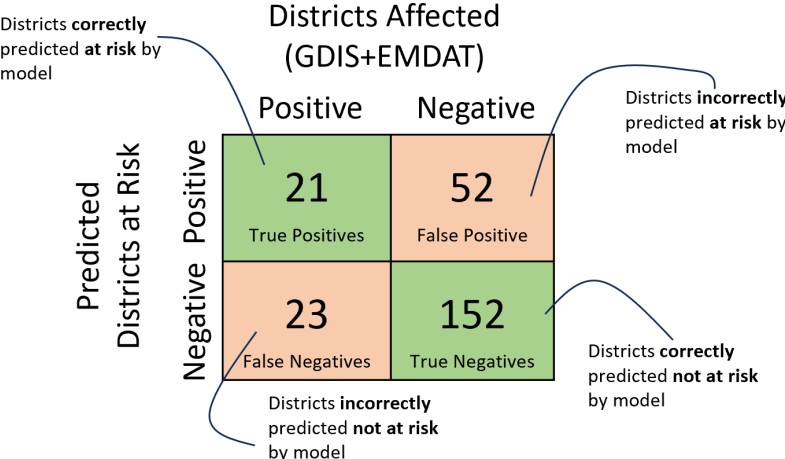

**Figure 7: Schematic diagram explaining the four components of the confusion matrix i.e., True Positives, False Positives, True Negative and True Negatives.**

## 4    Results

### 4.1    Flood Conditioning Factors

To generate comprehensive flood risk maps of the Ganga Basin, we first computed a Flood Hazard Index (FHI) by integrating eight hazard parameters namely elevation, slope, flow accumulation, drainage density, distance to the river, distance from the confluence, rainfall and geomorphology using the AHP methodology (section 3.3). The FHI represents the probability of the location suffering from a potential flood based on hydrological and geomorphic factors in the study region (Mishra and Sinha, 2020). Supplementary Table 1 shows the decision matrix of the chosen eight parameters (level 1 hierarchy) and their relative weights, showing a consistency ratio (CR) for all the decision matrix tables was satisfactory as CR <= 0.10 and statistically validated our decision-making process. Similarly, the decision matrix for each parameter was constructed and the relative of sub-factors were computed. Supplementary Table 2 to Supplementary Table 9, shows the consistency ratio (CR) for all the decision matrix tables was satisfactory as CR <= 0.10 and statistically validated our decision-making process. A summary of the relative weights of both the levels of hierarchy and the derived normalised weight is shown in Table 4.

Elevation is the primary or the most important parameter for assessing potential flood hazard having a level 1 relative weight (RW) of ca. 0.31 or 31% (Table 4). Elevation in the region ranges from 0-8000m, which has been categorised into five categories based on geometric classification - 0-115, 115-130, 130-250, 250-1,115 and 1,115-8,000 meters respectively. This corresponds to the relative weights of ca. 49%, 27%, 14%, 6% and 4% respectively (Table 4). This is because the low-lying areas are clearly more prone to floods due to water inundation. Thus, the low elevation (or flat areas) tend to have a higher level 2 relative weightage of ca. 49% compared to higher elevation areas having a lower RW of ca. 4%.

Similarly, slope is the second most important criterion for flood hazard assessment having a level RW of ca. 0.21 or 21%. In the study area, the slope is categorized into five zones geometric classification, i.e., 0∘-1.5∘, 1.5∘-1.7∘, 1.7∘-3.2∘, 3.2∘-13∘ and 13∘-90∘ respectively. This corresponds to the relative weights of ca. 49%, 27%, 14%, 6% and 4% respectively (Table 4). Extensive flat land with a very gentle slope suffers from prolonged inundation, whereas moderate to high slope provides an easy passage to pass away the flood water (Ghosh and Kar 2018). Additionally, the topographic gradient has considerable influence over the infiltration rate (Das 2020). Therefore, large volumes of water become sluggish near the sites of low-lying



flat topography (Bui et al. 2019). Thus, a low geographic slope usually displays greater susceptibility to flood flat areas and
tends to have a higher level 2 relative weightage of ca. 49% compared to steep areas having a lower RW of ca. 4%.
Next, the intensity of rainfall is an important parameter for assessing flood risk, having a level 1 RW of ca. 0.15 or 15%. In
our study, rainfall is categorised into five classes of geometric classification - 0-200, 200-260, 260-290, 290-350, and >350
mm/month respectively. This corresponds to the relative weights of ca. 49%, 27%, 14%, 6% and 4% respectively (Table 4).
The amount of rainfall is usually directly proportional to the intensity of flooding, especially during the summer monsoon,
where major floods in the area are triggered by the rainfall which generates high surface runoff. Also, high sediment flux from
the catchment during the rainfall events and limited accommodation space in the channel encourage siltation and therefore
breaching and flooding (Mishra and Sinha 2020). Thus, areas experiencing high rainfall have a higher level 2 relative
weightage of ca. 49% compared to lower rainfall areas having a lower RW of ca. 4%.
As floods are frequent in areas closer to a river, distance to rivers is an important flood hazard factor having a level 1 RW of
ca. 0.12 or 12%. In the study, we divided the distance from the major rivers of the basin into seven categories, primarily 0-
500, 500-1000, 1000-1500, 1500-2000, 2000-2500, 2500-3000, and 3000-3500 meters respectively. This corresponds to the
relative weights of ca. 37%, 24%, 16%, 10%, 7%, 4% and 2% respectively. Distance from rivers is another significant factor
that plays a vital role in determining flood conditions. According to several researchers, (Rahmati et al. 2016; Ghosh and Kar,
2018; Bui et al, 2019) flooding is typically expected in the areas near the river due to heavy runoff in the drainage system,
mainly after intense rainfall, which results in the river exceeding the limit of stream capacity. Thus, areas in close proximity
to the river of 0-500m have a higher level 2 relative weightage of ca. 37% compared to areas away from the river having a
lower RW of ca. 2%
Similarly, areas near river confluences are often prone to flooding. In the study, we divided the distance from the confluence
points into five categories- 0-500, 500-1000, 1000-1500, 1500-2000 and 2000-3000 meters respectively. This corresponds to
the relative weights of ca. 49%, 27%, 14%, 6% and 4% respectively (Table 4). During the monsoon season, these rivers carry
large volumes of water, leading to fluctuations in water levels and flow velocities. Additionally, the high sediment load,
aggradation, and reduced conveyance capacity can become problematic, especially when a channel has to handle the combined
flow of multiple tributaries, particularly during floods. This often results in the overtopping of riverbanks inundating nearby
floodplains (Ghosh and Kar, 2018). Thus, areas within 0-500m of the confluence points have a higher level 2 relative weightage
of ca. 49% compared to areas 2000-300m away from the confluence points which has a lower RW of ca. 4%.
Next watershed morphometric parameters that are important for assessing flood hazard and flood risk are drainage density and
flow accumulation, having a level 1 RW of ca. 0.06 and 0.05 respectively. In the study, drainage density is classified into 5
categories using a geometric classification - 0-0.00002, 0.00002-0.000032, 0.000032-0.000052, 0.000052-0.000089, and
0.000089-0.000155 meters$^{-1}$ respectively. This corresponds to the relative weights of ca. 49%, 27%, 14%, 6% and 4%
respectively (Table 4). In general terms, high drainage density indicates a great amount of surface runoff (e.g. Das and Pardeshi
2018). Hence, the regions having a dense stream network generally show frequent flooding due to higher drainage density
(Ogden et al., 2011; Das, 2020; Roy et al., 2021). Thus, areas having high drainage density of range 0.000089-0.000155m$^{-1}$
have higher level 2 relative weightage of ca. 0.48 compared to low drainage density areas of range 0-0.00002 m$^{-1}$ having a
lower RW of ca. 0.04 (Table 4). Drainage density was followed by flow accumulation which signifies total flow to a particular
point within the catchment from upstream areas and thus, higher flow accumulation indicates a high possibility of flooding. It
is classified into 5 categories in the study geometric classification - 0-1, 1-2, 2-26, 26-314, and 314-3800 respectively.
Naturally, a greater amount of accumulated flow leads to increased runoff in a low-elevated area, which contributes to more
flooding. Thus, areas having a high accumulation of 314-3800 unit cells have a higher level 2 relative weightage of ca. 49%
compared to areas having a low flow accumulation of 0-1 unit cells which has a lower RW of ca. 4%.





The last parameter of importance is basin geomorphology having a level 1 RW of ca. 0.02 or 2%. It is important to highlight that the overall geomorphology of the Ganga basin does not influence or trigger the floods directly but the presence of individual geomorphic features influences the potential of flooding and duration of inundation in a given region. Therefore, geomorphology has been ranked the lowest in the hierarchy in comparison to the other factors (Mishra and Sinha, 2020). In our study based on the geomorphic features in the basin, the layer is divided into ten categories, namely - active floodplains, older floodplains, alluvial plains, water bodies, coastal, anthropogenic terrain, pediment-peneplain, plateau, hills and valleys, and aeolian plane, which corresponding have relative weights of 29%, 21%, 14%, 10%, 9%, 7%, 4%, 3%, 2% and 1% respectively. Among all the geomorphic units, active floodplains are most prone to frequent flooding compared to older floodplains. Thus, active floodplain areas have a higher level 2 relative weightage of ca. 29% compared to the aeolian plane having a lower RW of ca. 1%.

**Table 4: Flood Hazard Index (where RW- relative weight)**

| Parameters | | Level 1 RW | Categories | Level 2 RW | Flood Hazard Index (Level 1*2) | Percentage (%) |
|---|---|---|---|---|---|---|
| | | | | | | |
| Elevation (in meters) | E | 0.31 | 0-115 | 0.49 | **0.150389** | 15.04 |
| | | | 115-130 | 0.27 | **0.081937** | 8.19 |
| | | | 130-250 | 0.14 | **0.041752** | 4.18 |
| | | | 250-1,115 | 0.06 | **0.020483** | 2.05 |
| | | | 1,115-8,000 | 0.04 | **0.013643** | 1.36 |
| | | | | | | |
| Slope (in degrees) | S | 0.21 | 0-1.5 | 0.49 | **0.10105** | 10.10 |
| | | | 1.5-1.7 | 0.27 | **0.055055** | 5.51 |
| | | | 1.7-3.2 | 0.14 | **0.028054** | 2.81 |
| | | | 3.2-13 | 0.06 | **0.013763** | 1.38 |
| | | | 13-90 | 0.04 | **0.009167** | 0.92 |
| | | | | | | |
| Rainfall (in mm/month) | R | 0.15 | >350 | 0.49 | **0.071491** | 7.15 |
| | | | 290-350 | 0.27 | **0.03895** | 3.90 |
| | | | 260-290 | 0.14 | **0.019848** | 1.98 |
| | | | 200-260 | 0.06 | **0.009737** | 0.97 |
| | | | 0-200 | 0.04 | **0.006485** | 0.65 |
| | | | | | | |
| Distance to the river (in meters) | DR | 0.12 | 0-500 | 0.37 | **0.045778** | 4.58 |
| | | | 500-1000 | 0.24 | **0.029748** | 2.97 |
| | | | 1000-1500 | 0.16 | **0.019787** | 1.98 |
| | | | 1500-2000 | 0.10 | **0.013019** | 1.30 |
| | | | 2000-2500 | 0.07 | **0.008697** | 0.87 |
| | | | 2500-3000 | 0.04 | **0.005092** | 0.51 |
| | | | 3000-3500 | 0.02 | **0.002895** | 0.29 |
| | | | | | | |
| Distance from river confluence (in meters) | DC | 0.08 | 0-500 | 0.49 | **0.040997** | 4.10 |
| | | | 500-1000 | 0.27 | **0.022336** | 2.23 |
| | | | 1000-1500 | 0.14 | **0.011382** | 1.14 |
| | | | 1500-2000 | 0.06 | **0.005584** | 0.56 |
| | | | 2000-3000 | 0.04 | **0.003719** | 0.37 |



| | | | | | | |
|---|---|---|---|---|---|---|
| | | | | | 0.49 | 0.0293 | 2.93 |
| Drainage Density (in meters$^{-1}$) | **DD** | **0.06** | 0.000089-0.000155 | 0.49 | **0.0293** | 2.93 |
| | | | 0.000052-0.000089 | 0.27 | **0.015963** | 1.60 |
| | | | 0.000032-0.000052 | 0.14 | **0.008134** | 0.81 |
| | | | 0.00002-0.000032 | 0.06 | **0.003991** | 0.40 |
| | | | 0-0.00002 | 0.04 | **0.002658** | 0.27 |
| | | | | | | |
| Flow Accumulation (unit cells) | **FA** | **0.05** | 314-3800 | 0.49 | **0.022133** | 2.21 |
| | | | 26.5-314 | 0.27 | **0.012059** | 1.21 |
| | | | 2.2-26.5 | 0.14 | **0.006145** | 0.61 |
| | | | 0.17-2.2 | 0.06 | **0.003014** | 0.30 |
| | | | 0-0.17 | 0.04 | **0.002008** | 0.20 |
| | | | | | | |
| Geomorphology | **G** | **0.02** | Active floodplains | 0.29 | **0.006895** | 0.69 |
| | | | Older floodplains | 0.21 | **0.005093** | 0.51 |
| | | | Alluvial Plains | 0.14 | **0.003331** | 0.33 |
| | | | Water Bodies | 0.10 | **0.002458** | 0.25 |
| | | | Coastal | 0.09 | **0.002037** | 0.20 |
| | | | Anthropogenic Terrain | 0.07 | **0.001575** | 0.16 |
| | | | Pediment-Peneplain | 0.04 | **0.000901** | 0.09 |
| | | | Plateau | 0.03 | **0.000626** | 0.06 |
| | | | Hills and Valleys | 0.02 | **0.000496** | 0.05 |
| | | | Aeolian Plane | 0.01 | **0.000346** | 0.03 |
| | | | | | | |
| | | | | | | 100.00 |




**4.2    Flood Hazard Assessment**

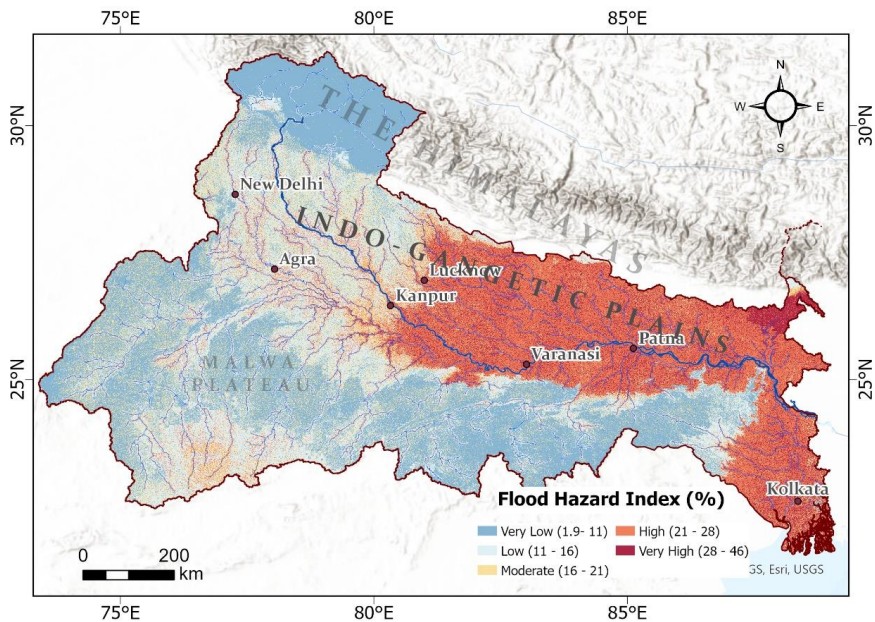


**Figure 8: Flood Hazard Index (FHI) map for 2022 of the Ganga Basin produced after the AHP analysis. Continuous data ranging**
**from 1.9 to 46% are classified into 5 categories: Very Low, Low, Moderate, High and Very High FHI. In the figure the blue areas**
**have low FHI values, indicating less potential for flood occurrence or flood hazard, and red areas have higher flood hazard values,**
**suggesting higher susceptibility to floods.**
To determine the flood hazard index (FHI), we integrated thematic layers related to flood factors. Using the relative weights
(RWs) from both level 1 and level 2 hierarchies provided in Table 4, we calculated the FHI for each pixel (i) from Equation 6
for the years 2013 through 2023. Supplementary Figure 1 shows the derived FHI maps from 2013 to 2023. The analysis reveals
that the FHI varies both spatially and temporally. The temporal variations in FHI are primarily influenced by annual rainfall,
which is treated separately each year. For clarity and to illustrate the spatial distribution of flood-prone areas, Figure 8 displays
the FHI map for 2022. This map shows FHI values ranging continuously from ca. 0.019 to 0.49 or 1.9% to approximately
46%, categorised into five levels: very low (1.9-11%), low (11-16%), moderate (16-21%), high (21-28%), and very high (28-
46%). Figure 8 indicates that flood risk is particularly high in the Indo-Gangetic plains, notably affecting major cities like
Kanpur, Lucknow, Varanasi, Patna, and Kolkata. In contrast, mountainous and arid regions such as the Himalayas, Malwa
Plateau, and the western part of the basin exhibit lower FHI values. This spatial variability is influenced by consistent
topographic factors, including elevation, slope, distance to rivers, distance from confluences, drainage density, flow
accumulation, and geomorphology, which remained constant throughout the analysis period.
**4.3    Flood Exposure Index (FEI)**
For assessing the exposure to flood hazard, the Flood Exposure Index (FEI) was determined across the Ganga basin, using the
annual night-time lights data from 2013 to 2023 and applying the Analytical Hierarch Process (AHP) method as outlined in
section 3.2. According to Supplementary Table 10, the radiance indicated by the night-time lights data ranges from 0 to 4000
$nW/cm^2sr^1$, which was classified into five categories based on geometric classification, i.e., 0-10, 10-17, 17-34, 34-293 and
>=293 $nW/cm^2sr^1$. Based on the decision matrix table constructed through AHP, the derived relative weights (RW) for 0-10,
10-17, 17-34, 34-293 and >=293 $nW/cm^2sr^1$ are ca. 0.04, 0.06, 0.14, 0.27 and 0.49 respectively. Our decision was statistically
validated by the consistency index (CI) and the consistency ratio (CR), where the CR for the exposure decision matrix table





was 0.051, which is below the acceptable threshold of 0.10 as shown in Supplementary Table 10. The calculated relative
weights correspond to the Flood Exposure Index (%). Supplementary Figure 2 illustrates the derived flood exposure index
(FEI) from 2013 to 2023 in the Ganga Basin. The brighter areas having higher intensity are indicative of high exposure
compared to the dark areas, having lower exposure in non-inhabited places such as forests. For detailed visualisation, Figure
9 displays the FEI map for 2022. Figure 9a highlights that areas with a very high FEI (around 49%) correspond to regions with
significant night-time lighting (radiance ≥293 nW/cm²sr), suggesting a dense presence of infrastructure. Conversely, areas
with a very low FEI (around 4%) reflect minimal night-time lighting (radiance 0-10 nW/cm²sr), indicating sparse infrastructure.
The spatial distribution of FEI indicates that the highest exposure is concentrated in the plains of the Ganga Basin, particularly
in regions stretching from New Delhi to Kolkata. This highlights the greater exposure of densely populated areas to flood
events due to the presence of human activities reflected by artificial lights. Meanwhile, southern and western parts of the study
area exhibit lower exposure levels. Figure 9b focuses on the northeastern part of the basin, particularly over the state of Bihar,
where the FEI map reveals clusters of lit infrastructure near the Kosi and Ganga rivers, indicating higher flood exposure in
these regions. These results demonstrate the higher importance of areas with large human presence compared to low-exposure
areas for prioritizing flood risk efforts.

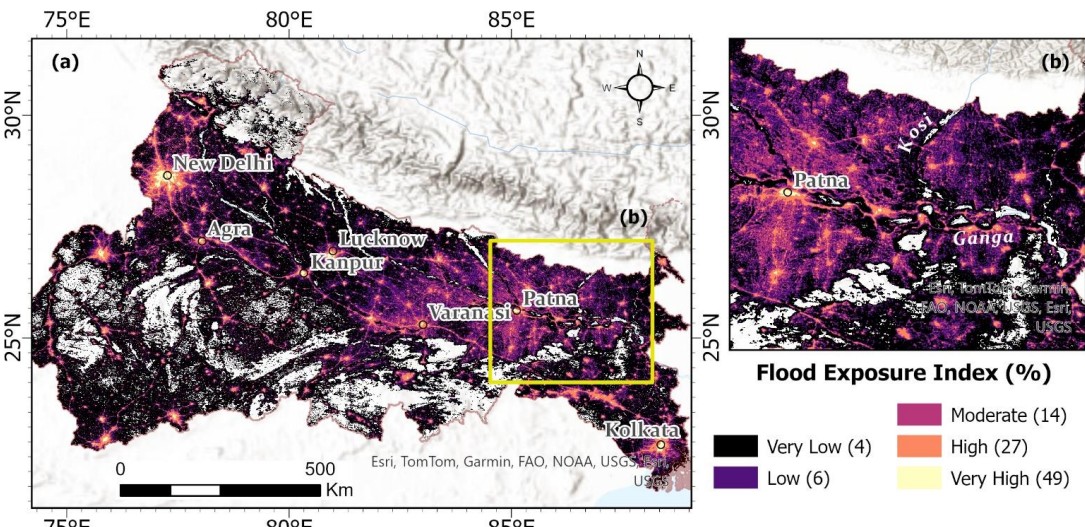


**Figure 9: Flood Exposure Index (FEI) map for 2022 across the Ganga Basin produced after the AHP. The FEI is categorized into 5**
**classes based on the unique values of weights derived, which are: Very Low (4%), Low (6%), Moderate (14%), High (27%) and**
**Very High (49%).**
**4.4  Flood Vulnerability Index (FVI)**
In this study, the Flood Vulnerability Index (FVI) for the Ganga basin was determined using the district-wise vulnerability
index from the Climate Vulnerability Report 2020 and the Analytic Hierarchy Process (AHP) method. The continuous index
was divided into five categories, ranging from low (0.35-0.425) to very high (0.65-0.73) based on the report classification, as
outlined in Supplementary Table 11. The AHP method allowed us to derive the relative weights (RW) for each category, with
Supplementary Table 11 providing a detailed decision matrix. The calculated RWs for the five categories were approximately
0.04, 0.06, 0.14, 0.27, and 0.49, respectively. This analysis showed that areas with higher vulnerability (0.65-0.73) exhibited
a significantly larger RW of around 0.49, indicating that nearly 49% of the vulnerability can be attributed to the factors
considered in this range. In contrast, areas with lower vulnerability values (0.35-0.425) were associated with a considerably
smaller RW of approximately 0.04, reflecting a lower contribution to the overall FVI. After computing the decision matrix for



the five vulnerability classes, we conducted a consistency check to ensure the robustness of the AHP model. The Consistency
Index (CI) and Consistency Ratio (CR) were calculated for the decision matrix. The CR value for the exposure decision matrix
was found to be 0.051, which is well within the acceptable threshold of 0.10, indicating a satisfactory level of consistency in
the judgments made during the weighting process. This result underscores the validity of the AHP-derived weights and
confirms that the classification scheme is statistically sound.
The higher RW for areas in the 0.65-0.73 range highlights the critical need for targeted flood mitigation efforts in these regions,
as they are the most vulnerable based on the selected indicators. Similarly, areas in the lower vulnerability category, although
less at risk, should still be monitored, given their susceptibility to changes in biophysical, socio-economic, and infrastructural
conditions. Additionally, spatial mapping of the FVI across the basin in Figure 10 reveals clear geographic patterns of
vulnerability. The northwestern and central regions of the Ganga basin show a higher concentration of districts falling into the
high and very high vulnerability categories, while the southeastern and northern regions are generally associated with lower
vulnerability. This spatial distribution suggests the focus of vulnerability in assessing flood risk within the basin. This should
help to provide a clear framework for prioritising flood mitigation resources and policy interventions in the most at-risk areas
within the basin.
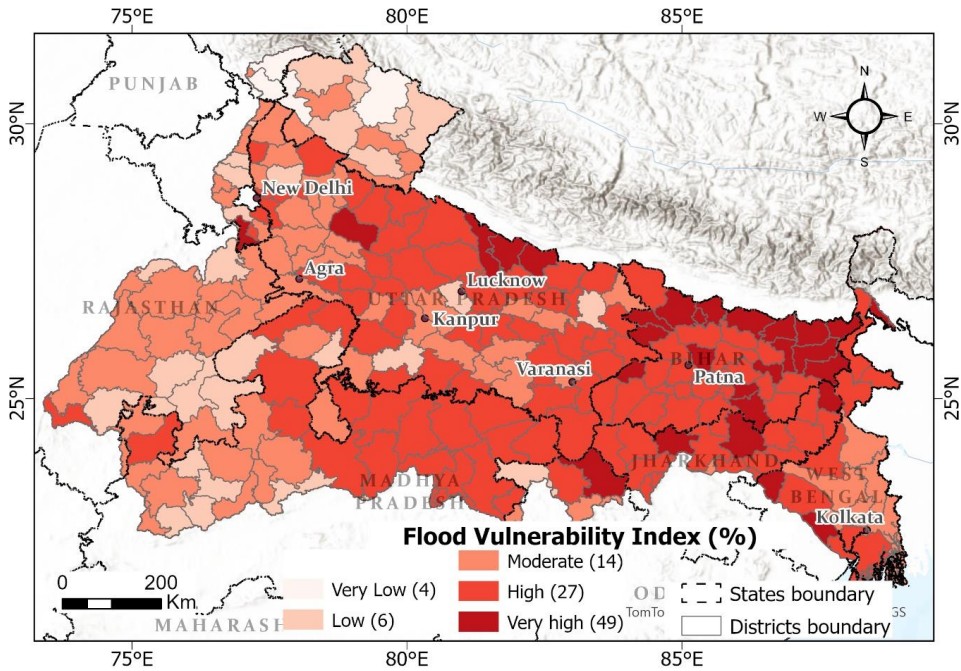
**Figure 10: Flood Vulnerability Index (FVI) map for 2022 across the Ganga Basin produced after the AHP analysis. The FVI is**
**categorized into 5 classes based on the unique values of weights derived, which are: Very Low (4%), Low (6%), Moderate (14%),**
**High (27%) and Very High (49%).**
**4.5    Flood Risk Map**
The annual normalized Flood Risk Index (FRI) map of the Ganga Basin from 2013 to 2023 is calculated at each pixel using
Equation 7, which combines the Flood Hazard Index (FHI), Flood Exposure Index (FEI), and Flood Vulnerability Index (FVI).
The FRI values for this period range between 0 and 0.085. For better visualization and discussion, these original values are
normalized to a maximum of 0.09 and expressed as percentages in Supplementary Figure 3. Based on geometric classification,
the FRI percentages are divided into five categories: very low risk (0-1%), low risk (1-3%), moderate risk (3-10%), high risk
(10-30%), and very high risk (30-100%). Very low and low-risk categories are not presented in the results as they have minimal




or no significant contribution to overall flood risk. The maps demonstrate a temporal increase in flood risk distribution, with a
noticeable concentration in the eastern part of the basin, particularly over eastern Uttar Pradesh, Bihar, and West Bengal.
To illustrate spatial variability, the flood risk map for 2022 is presented as an example in Figure 11. In Figure 11a, the
distribution of moderate to very high flood risk pixels across the Ganga Basin is shown, with a significant concentration
towards the eastern region. Areas located farther from the river in the southwestern basin, including parts of Madhya Pradesh,
Rajasthan, and the northern mountainous region, exhibit minimal to no flood risk. Figures 11b- 11d offer closer views of the
flood risk in Bihar, Uttar Pradesh, and West Bengal. In Figure 11b, a large portion of Bihar shows moderate flood risk, with
the eastern areas experiencing high to very high risk. Similarly, Figure 11c highlights that the high-risk regions in Uttar Pradesh
are situated away from major urban centers such as Lucknow and Kanpur, emphasizing that flood risk is not solely determined
by exposure but reflects the comprehensive methodology applied in this study. Figure 11d shows the high-risk areas in West
Bengal, particularly in the northern and northwestern regions of the state. These findings provide a detailed understanding of
flood risk at a higher spatial resolution, which is influenced not just by individual flood potential hazards, exposure or
vulnerabilities but an amalgamation of all the aforementioned. However, we recognize the potential inconsistencies in the
classification into five risk categories, thus we calculated flood risk trends at each raster pixel to identify areas with increasing
flood risk trends which will be discussed in the section 5.1.

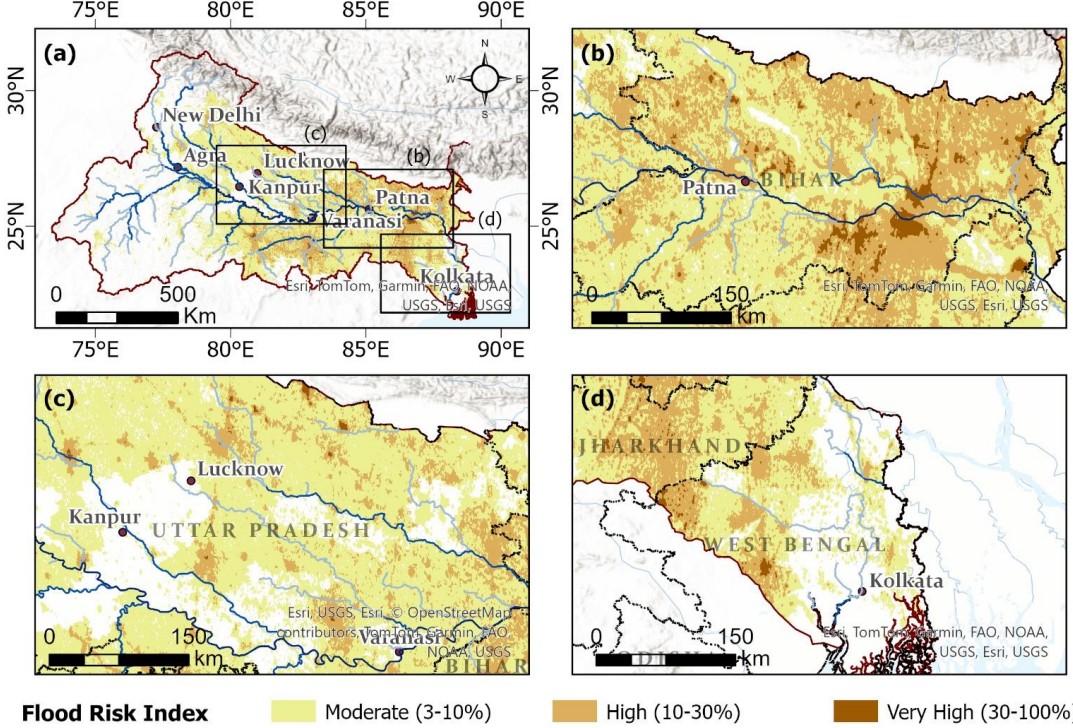


**Figure 11: (a) Normalised Flood risk index map of the Ganga Basin for 2022, highlighting areas of moderate to high risk. Yellow**
**indicates moderate risk (3-10%), orange represents high risk (10-30%), and brown denotes very high risk (30-100%). (b), (c), and**
**(d) provide zoomed-in views of specific geographic regions.**

**4.6    Validation**
Validating a model entails systematically comparing its outputs to real-world observations, and gauging prediction accuracy
in quantity and quality. Ensuring reliability for future flood risk evaluations demands validation on, aligning model outputs
with observed or ground truth data through calibration. For this study, the 2016 flood risk map was validated by comparing its




output with the flood impact inventory developed for the basin using the EM-DAT and GDIS data using the ROC-AUC
(Receiver Operating Characteristics-Area Under the Curve) method which is previously discussed in section 3.7. We chose
the 2016 specific case study because of 2 primary reasons. First, the common availability of years between the GDIS and the
risk map generated is between 2013 and 2018. Secondly, as per the Global Flood Database, the flood event triggered by heavy
rains on 25th July 2016 had the highest exposed population of ca. 18.5 million people between 2013 to 2018 in the Ganga Basin
(Tellman et al., 2021).
The ROC-AUC curve assesses the performance of the flood risk model by plotting the true positive rate against the false
positive rate. An AUC value signifies the model's ability to discriminate between actual flooded districts (derived from
EMDAT and GDIS) and flood risk districts from the model, with higher values indicating better accuracy. This validation
metric confirms the robustness of our flood risk map, providing a quantitative measure of its reliability in predicting and
classifying flood risk in the studied region. In Figure 12 we observe that the ROC-AUC curve plotted between the true positive
rate (TPR) and false positive rate (FPR) has an area under the curve (AUC) value of 0.69 (or 69%), which is satisfactory –
very good range as per the classification Das, 2020; Lin et al., 2019; Mukhtar et al., 2024; Roy et al., 2021; Saha and Agrawal,
2020). The model can discriminate and predict accurately if the AUC value is closer to 1.

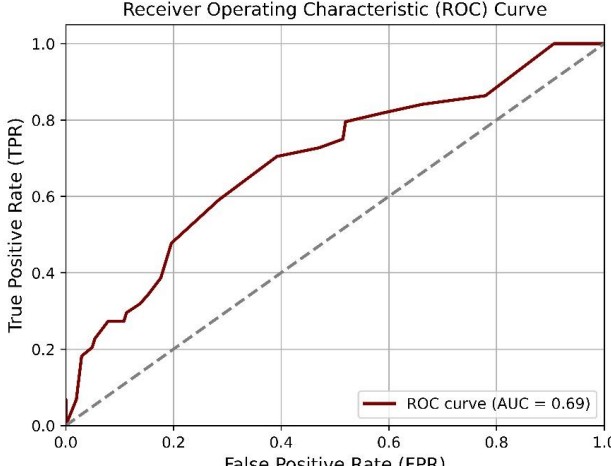


**Figure 12: ROC curve (maroon colour) plotted between the false positive rate and the true positive rates at different risk threshold**
**values. The AUC is the value of the area under the curve of the 2016 flood risk map.**
**5    Discussion**
**5.1    Comprehensive approach for identifying and predicting flood risk**
Our study adopts a GIS-based Analytical Hierarchy Process (AHP) method to characterise flood risk in the Ganga Basin, India.
The main objective was to integrate flood hazard, exposure, and vulnerability datasets while also demonstrating the utility of
the night-time lights (NTL) data for assessing flood risk. Numerous studies have suggested that the increase in risk in recent
years primarily due to the increase in exposure of assets and people in hazard-prone areas, with only minor increases in hazard
itself over the last decade (IPCC, 2012; Blaikie et al., 2014; Kundzewicz et al., 2014; Munich Re, 2014; Visser et al., 2014;
Jongman et al., 2015; GFDRR, 2016, de Ruiter et al., 2017; UNDRR, 2020). This growing risk is also linked to increased
vulnerability, a critical factor in flood damage assessments (Connor and Hiroki, 2005, de Ruiter et al., 2017; Mishra and Sinha,
2020). Our study allows us to address the changing flood hazard and exposure over time within the Ganga Basin. Flood hazards
in this basin are influenced by elevation, slope, flow accumulation, drainage density, proximity to rivers and confluences,



rainfall, and geomorphology. The impact of these hazards is amplified by the exposure and vulnerability to floods in various
areas. As discussed in the section 4.2 the majority of flood hazard factors have remained constant over the past 10 years
except for rainfall, which contributes to the temporal variation of flood hazard within the basin as seen in the Supplementary
Figure 1.
Our flood risk map of the Ganga Basin, developed using an integrated hydro-geomorphological approach with AHP,
outperforms traditional hydrological and hydraulic models as it combines the physical (geomorphological) criteria with hydro-
meteorological data. This emphasizes process-based understanding and overcomes the necessity of dense hydrological data as
required by the hydraulic modelling of floods (Mishra and Sinha, 2020). Our hazard map reveals that the majority of high
flood hazards in the basin are concentrated within the plains of the Ganga Basin and is majorly influenced by the elevation
(31%), slope (21%), rainfall (15%) and distance from the river (12%). Monsoonal precipitation was also identified as a key
factor, making it integral to the flood hazard index. However, the geomorphology had the least influence on the occurrence of
identifying flood-prone areas about 2%. Our model utilizes multiple data like the DEM, night-lights, and vulnerability index,
for understanding and predicting risk had an accuracy of ca.70% which is satisfactory (Das, 2020; Lin et al., 2019; Mukhtar et
al., 2024; Roy et al., 2021; Saha and Agrawal, 2020). Hazards that occur in areas with low vulnerability will not become
disasters (Quarantelli, 1998; Birkmann et al., 2013). However, most hazards occur in low and middle-income countries (LMIC)
with a high population density, poor infrastructure, and limited or no disaster preparedness plan (Mishra and Sinha, 2020). For
instance in Figure 8, the eastern part of the basin has very high FHI but it does not have a similar concentration of a very high
FRI in Figure 11. This highlights that the high flood hazard index does not necessarily relate to high flood risk as well,
highlighting the analysis of vulnerability and exposure to disasters, magnitude, and their impact (de Ruiter et al., 2017, Mishra
and Sinha, 2020).
To gain an understanding of the distribution of flood risk and identify areas with persistent or increasing risk, we derived the
trend of the flood risk from the flood risk maps from the last ten years, which is representative of percentage point change per
year. Figure 13a shows maps that highlight areas of increasing flood risk (red colours), with a noticeable concentration in the
eastern part of the basin, particularly in the state of Bihar, eastern parts of Uttar Pradesh, and Madhya Pradesh. Bihar has
periodically experienced floods due to extreme rainfall-induced riverine flooding, as several rivers flow in this area. Some of
the major flood events in the northern part of Bihar occurred in the years 1987, 1998, 2000, 2001, 2003, 2004, 2008, 2010,
2013, 2017, 2018 and 2020 (Tripathi et al., 2022).  Figure 13b, for example, illustrates an increasing trend of flood risk near
the confluence of the Ganga and Kosi rivers in Bihar. Moreover, areas along the current path of the Kosi River show a more
pronounced trend compared to regions farther from the river, particularly areas within and adjacent to the districts of Supaul,
Saharsa, and Khagaria. This elevated risk could be attributed to the Kosi River's historical avulsions, having shifted its course
westward by approximately 150 km over the past 200 years (Sinha, 2009; Mishra and Sinha, 2020). Additionally, the
Himalayan foothill regions have experienced regular flooding resulting in the loss of human lives and infrastructure damage
(Roy et al., 2021). Ooccasional landslides in the upper catchment areas of the rivers trigger sudden water release, causing
rivers to discharge massive volumes of water (Roy, 2011). Figure 13c displays a heightened flood risk trend near the foothill
regions of the Himalayas, particularly around Jamunaha. As the country's population has grown, settlements have moved to
the riverbanks, increasing the risk of erosion and flooding (Islam et al., 2023). Figure 13d shows that downstream areas of the
Ganga River in West Bengal exhibit varied trends, with the northern part facing higher flood risk compared to the southern
areas, where risk is either stable or decreasing. This is because the northern region possesses unique physiographic and
orographic conditions influenced by the eastern Himalayas, where numerous streams converge at the foothills, forming large,
braided rivers (Chakraborty and Mukhopadhyay, 2019). Finally, Figure 13e indicates that areas within and adjacent to the
districts of Satna and Rewa have a higher risk trend compared to other parts of the Madhya Pradesh state. This is due to the





variability of the southwest monsoon, which causes high rainfall in the eastern part of Madhya Pradesh and reduces towards
the north and western parts (Tiwari et al., 2024).

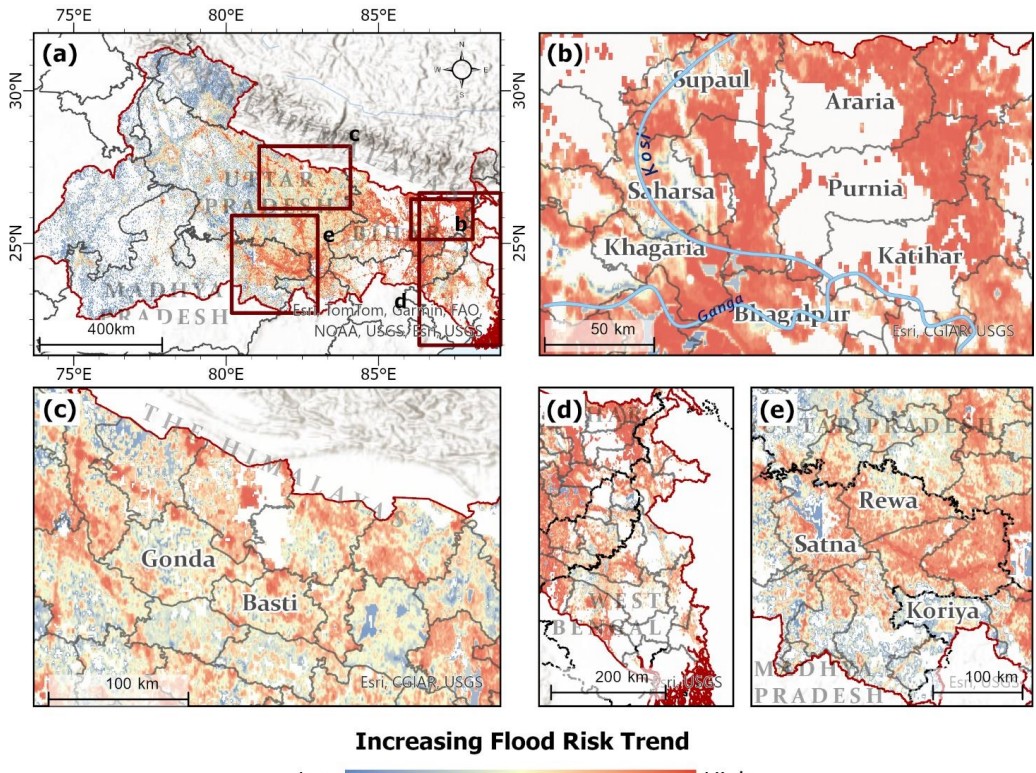


**Figure 13: (a) The spatial distribution of increasing flood risk trend within the Ganga Basin in India over the past 10 years. Each**
**pixel represents the average percentage point change in FRI per year, enabling us to identify areas of significantly increasing flood**
**risk. The blue areas represent regions with a lower rate of flood risk increase ca. 1 percentage point change/per year, while the red**
**areas indicate regions with a higher flood risk trend ca. 5 percentage point change/ per year; (b) shows a focused view of the trend**
**within the state of Bihar; (c) highlights the trend near the foothills of the Himalayas; (d) zooms in on the trend in the state of West**
**Bengal; (e) provides a detailed view of the trend over the state of Madhya Pradesh within the basin**.
The occurrences of episodic flooding over time, especially during the summer monsoon are a potential flood hazard which
becomes a disaster if leading to human and resource loss (UNDRR, 2020). The cumulative reasons for the exponential loss in
the area are the geomorphic factors, the exposed human population and the ability to cope with the floods. The key highlight
of our work is leveraging the temporal and spatial resolution of night-time lights as a proxy for exposure to flood hazard. In
general, the lack of data especially up-to-date or recent datasets provides a hindrance to robust estimation of flood exposure
and risk. As observed in Supplementary Figure 2 the distribution of lit pixels serving as a proxy for human exposure has
increased over the last decade, and consequently, we suggest that flood models should make use of recent datasets such as this,
representing population and human infrastructure. Compared with traditional statistical and census data, NTL can directly
reflect the real-time distribution of human activities, and address the issue of not being able to document human presence in a
timely manner using traditional methods (Fang et al., 2021).
Flood risk, which combines hazard, exposure, and vulnerability, is crucial for prioritising resources and interventions, as it
reveals the spatial distribution of potential damage (Elshorbagy et al., 2017). Thus the Analytical Hierarchy Process (AHP)
framework integrated with the GIS datasets and most importantly human population proxy datasets aids in formulating and
identifying flood risk areas. This study gives an estimate of the location at a high spatial resolution where the flood risk is high





and can be used in government policy implementation to minimise the potential flood damage. This kind of research can help
contribute to informed disaster planning for government and non-governmental organizations. For planners and decision-
makers, these maps therefore aid in evaluating areas for future development or strategic infrastructure.

For this approach to be of most value to stakeholders, it is important to address study limitations and areas for future refinement as this is essential for ensuring transparency regarding potential weaknesses and constraints within the research methodology. By acknowledging these limitations, the credibility of the findings is strengthened, offering a more informed interpretation of the results. This also paves the way for refining future research, contributing to a balanced and realistic assessment of the study's scope and implications. The AHP method relies on prior subjective knowledge and assumptions, leading to uncertainties, but its main advantage is that it does not require historical datasets. In contrast, machine learning (ML) models based on historical data are rapidly evolving, and they can improve model accuracy by incorporating multiple parameters for multidimensional analysis of the significance or weighting of different factors (Khosravi et al., 2019). The study assumes a constant vulnerability level over the past decade for simplification purposes, reasoning that vulnerability components wouldn't have changed significantly. However, given rising exposure, future work should account for variations in vulnerability over the decade. Night-time data was used to assess exposure, as it effectively captures the spatial distribution of human infrastructure. While useful for disaster assessment, its spatial resolution is limited to 500 meters which is a limitation for neighbourhood-specific flood risk approaches. The validation process was based on disaster impact data repository from EM-DAT and GDIS data, two of the most comprehensive global disaster databases. While these sources are verified, data gaps stem from limited capacity and resources to fully document all events. Despite these limitations, we opted for this data due to the lack of alternative impact-based datasets, although they may have accuracy constraints and may not offer the most optimal validation framework. Additionally, using higher-resolution data could improve the precision and robustness of the study's findings.

## 6 Conclusion

Most previous research on flood risk has primarily relied on exposure data, such as population or infrastructure, especially in developing regions where data availability is limited. The novelty of the work lies in using night-time lights as a proxy for exposure within the basin, unlike the conventional population data. This study leverages the temporal availability of the data, enabling a real-time distribution of human activities at a large scale and with greater temporal resolution. Unlike some earlier studies that used the terms hazard, exposure, and vulnerability interchangeably to assess flood susceptibility, this research treats these components separately to better understand flood risk in one of the world's most densely populated basins—the Ganga Basin. In this basin, the Indian summer monsoon between June to September contributes about 80% of the yearly rainfall, affecting about 18 million people, with an increased frequency and magnitude of flooding being a major concern in recent years. In this research through a comprehensive literature review, we identified the key drivers of the flood and their relative weights using the Analytical Hierarchy Process. These drivers include elevation (31%), slope (21%), rainfall (15%) distance from the river (12%), distance from the confluence (8%), flow accumulation (6%), drainage density (5%) and geomorphology (2%). Using these data, the night-time lights and the vulnerability data from the report, we computed the flood hazard index (FHI), flood exposure index (FEI) and flood vulnerability index (FVI) to assess how the flood risk has evolved over the past decade from 2013 to 2023. The findings indicated certain areas had high FHI meaning higher susceptibility to floods but did not have high flood risk. Our analysis based on the flood risk trend reveals that there is a significant increase in flood risk trend in the eastern part of the basin, particularly areas in Bihar, eastern Madhya Pradesh, eastern Uttar Pradesh and the northern part of West Bengal, identifying high flood risk zones at the pixel or cell level. Conversely, a lower risk trend is observed in other parts of the basin, which can be attributed to variations in the southwest summer monsoon and expanding human exposure in certain areas. This research represents a first-of-its-kind effort to utilise nighttime lights data to examine





the temporal dynamics of flood risk, providing a novel approach for assessing and tracking increasing flood risk in the
region. A better understanding of evolving hazards and exposure is crucial in areas having limited or outdated population
information, informing policymakers as it allows them to identify areas that are experiencing increasing flood risk.
**Code Availability**
The code for processing the processing the raster data and computation of flood risk index is available upon request.
**Data Availability:**
The ASTER Digital Elevation Model (DEM) is available https://appeears.earthdatacloud.nasa.gov/; Climate Hazards Group
InfraRed Precipitation with Station data (CHIRPS) is available at https://developers.google.com/earth-
engine/datasets/catalog/UCSB-CHG_CHIRPS_DAILY#description; The Geological Survey of India geomorphology data is
available at https://bhukosh.gsi.gov.in/Bhukosh/Public; The NASA Black Marble nighttime lights annual product suite
(VNP46) is available at https://ladsweb.modaps.eosdis.nasa.gov/search/order/2/VNP46A4--5000; The Climate Vulnerability
Assessment Report- DST 2020, Government of India is available at https://dst.gov.in/national-climate-vulnerability-
assessment-identifies-eight-eastern-states-highly-vulnerable; The EM-DAT data is available at https://public.emdat.be/ and
the GDIS is available at https://sedac.ciesin.columbia.edu/data/set/pend-gdis-1960-2018.
**Author Contributions:**
**Ekta Aggarwal** – conceptualization, methodology, software, validation, formal analysis, data curation, writing—original draft
preparation, writing—review and editing, visualization; **Marleen C. de Ruiter** – conceptualization, methodology,
investigation, validation, writing- review and editing, visualization, supervision; **Kartikeya S. Sangwan** - methodology,
investigation, writing- review and editing; **Rajiv Sinha –** methodology, formal analysis, writing- review and editing; **Sophie**
**Buijs** – methodology, writing – review and editing; **Ranjay Shreshtha** – methodology, writing – review and editing; **Sanjeev**
**Gupta** – writing – review and editing, supervision, funding acquisition; **Alexander C. Whittaker** – Conceptualization,
methodology, formal analysis, writing – review and editing, visualization, supervision, project administration, funding
acquisition.
**Competing interests:** The contact author has declared that none of the authors has any competing interests
**Funding:** This research was funded by the European Union's Horizon 2020 research and innovation program under the Marie
Sklodowska-Curie grant agreement No 86038. MCdR received support from the MYRIAD-EU project, which received
funding from the European Union's Horizon 2020 research and innovation programme under grant agreement No. 101003276.
MCdR also received support from the Netherlands Organisation for Scientific Research (NWO) (VENI; grant no.
VI.Veni.222.169).

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
