# Peer review of "Increasing flood risk in the Indian Ganga Basin: A perspective from"

_EGUsphere, 2024_

## Referee Comment (RC3)

**Title: Increasing flood risk in the Indian Ganga Basin: A perspective from the night-time lights**

This study assessed the flood hazard, exposure, vulnerability and risk level of Indian Ganga Basin based on a multi-criteria risk assessment methodology (hierarchical analysis) using selected hazard, exposure and vulnerability assessment indicators. The novelty of the study lies in using night-time lights as a proxy for exposure within the basin, unlike the conventional population data.

I have a few comments/questions:

(1) The abstract section of the manuscript is illogical and lengthy. For example, the innovation of the paper is highlighted twice in the beginning and in the end of the abstract. It is recommended that the authors rewrite the abstract section according to research background, methods and content, results, validation, innovativeness and application value.

(2) The confusing use of professional terms such as "flood risk" and "flood susceptibility" means that the professional level of research manuscripts needs to be improved.

(3) I disagree with the discussion of the superiority of the methodology of this study in lines 652-655 of the manuscript. Accurate flood risk assessment results are obtained by driving a hydrological-hydraulic model to capture flood hazard, and then combining exposure, vulnerability, and level of prevention and mitigation. The use of multi-criteria methods to assess flood risk levels in this study may be limited by data and technical expertise. Hydrological and hydrodynamic models can simulate the depth and range of flooding and obtain more accurate risk assessment results.

Lines 652-655: "Our flood risk map of the Ganges basin was developed using an integrated hydrogeological approach and AHP methodology, which is superior to traditional hydrological and hydraulic modelling as it combines physical (geomorphological) criteria with hydrometeorological data. This emphasises a process-based understanding that overcomes the need for intensive hydrological data required for flood hydraulic models (Mishra and Sinha, 2020)".

(4) The vulnerability assessment in the study used existing vulnerability products, the reliability of which was not verified. In addition, which indicators were considered in the vulnerability assessment were not clearly given.

(5) The novelty of the work lies in using night-time lights as a proxy for exposure within the basin, unlike the conventional population data. However, in the flood risk assessment framework based on multi-criteria methods, using night light data instead of population and economic data as an innovation in flood risk assessment seems to be somewhat insufficient for the entire study, but it is acceptable.

In flood risk assessment, research innovation would be more prominent if hazard assessment used flood inundation information obtained based on remote sensing data while taking into account changes in flood vulnerability.

(6) The dynamic trends of flood risk levels should be placed in the results section of the study rather than the discussion section, which should focus on the superiority of using nighttime light data for flood exposure and risk assessment.

---

## Author Comment (AC1)

**RC 1: (posted on 3rd April 2025)**

This study evaluates flood risk in the Ganga River Basin using the Analytical Hierarchy Process (AHP) approach, considering risk as a function of hazard, exposure, and vulnerability. A key aspect of the study is the use of night-time lights as a proxy for flood exposure, which is presented as a novel contribution. However, I find the study lacking in terms of scientific innovation, methodological advancements, and practical applications. My primary concern is the reliance on proxy variables for flood hazard assessment rather than actual hazard data. Detailed comments are as follows:

- *Lack of novelty:* AHP is a widely used method in flood risk assessments across various regions, including India. A simple literature search reveals numerous similar studies applying AHP in flood risk analysis…The present study primarily replicates an established approach with minor variations in proxy variables, offering limited scientific advancement:

  https://doi.org/10.1007/s11600-018-0233-z

  https://doi.org/10.1007/s10661-022-10111-x

  https://doi.org/10.1007/s11069-018-3392-y

  https://doi.org/10.1007/s11069-019-03737-7

  https://doi.org/10.5194/hess-21-2219-2017

  https://doi.org/10.1186/s40677-016-0044-y

  https://doi.org/10.1007/s12524-008-0034-y

  We thank the reviewer for pointing to relevant literature that indeed confirms the wide applicability and acceptance of the AHP method in flood risk assessments. While AHP has been employed extensively in similar contexts **the primary novelty and importance of our study lies not in the methodological framework itself, but in the integration of a dynamic exposure dataset—specifically, night-time lights (NTL)**. We do agree with the reviewer that we should have explained this better in the introduction. The purpose of employing AHP in this study was intentional, as it is a tested and interpretable method. Rather than replacing the methodology, our goal was to **strengthen it with contemporary data inputs** that can improve its responsiveness and relevance. This aligns with scientific practice where methodological frameworks are enhanced through novel inputs or contextual applications. The review recognises the NTL work is a novel contribution, and we believe this integration represents an important advancement in data-driven flood risk assessment for several key reasons:

  1. **NTL as an Exposure Proxy:** While previous studies (e.g., Gosh and Kar, 2018 and Danumah et al., 2016) rely on static or census-based demographic and infrastructure indicators, our study uniquely demonstrates the use of night-time light intensity as a spatially and temporally continuous proxy for human exposure and urbanisation. NTL has not to date been incorporated within flood risk literature and offers high-resolution insights into evolving exposure patterns.
  2. **Temporal Scope and Resolution:** The use of annual NTL data over a 10-year period introduces a longitudinal dimension to exposure analysis, capturing growth of human presence dynamics that are often missed in single-year assessments, which is often the case in using the

census-based population dataset. This enhances the capacity of the AHP model to reflect exposure changes—an important improvement over conventional static dataset.

3. **Enhanced Decision-Making Utility:** By integrating a globally available, regularly updated dataset (NTL), the approach we present can be more easily replicated across data-scarce regions. It aligns with the goal of developing scalable, low-cost methodologies for flood risk assessment, especially relevant for rapidly urbanising regions in the Global South.

Although the core methodological approach is established, our work provides added value by operationalising the NTL dataset in a practical, geospatial, and decision-support context—an aspect that is not well addressed in the cited literature. To make this as clear as possible for readers revised the manuscript to more explicitly emphasise the contribution of the study in terms of data innovation within a validated analytical framework.

We have restructured the Introduction section, where we have addressed the above clarifications, and emphasised the importance of NTL.

Line 83-94 reads as *"The key novelty of the approach lies in using night-time lights (NTL) as a proxy for flood exposure within the basin, unlike the population data, and leveraging the temporal availability of the data. NTL data is collected from satellite-based sensors like VIIRS (Visible Infrared Imaging Radiometer Suite) which offers a dynamic, consistent, and spatially explicit view of human activity and settlement patterns. Over the past decade, NTL data have been increasingly used to monitor urban expansion, economic activity, and disaster impacts (Andries et al., 2023; Román et al., 2018; Wang et al., 2018). The NTL data can reflect the real-time distribution of human activities at a large scale and with better temporal frequency, compared to traditional statistics and census data (Fang et al., 2021). More recently, studies have explored its application to examine human exposure and presence near rivers, including those associated with floods (Aggarwal et al., 2024; Ceola et al., 2014). Elshorbagy (2017) prepared flood exposure map of Canada is developed using a land-use map and the satellite-based nightlight luminosity data as two exposure parameters. The use of annual NTL data for flood risk assessment over a 10-year period introduces a longitudinal dimension to exposure analysis, capturing growth of human presence dynamics that are often missed in single-year assessments, which is often the case in using the census-based dataset."*

Line 151 – 154: *"The purpose of employing AHP in this study is to strengthen the framework with contemporary data inputs that can improve its responsiveness and relevance. This aligns with scientific practice where methodological frameworks are enhanced through novel inputs or contextual applications. This enhances the capacity of the AHP model to reflect exposure changes—an important improvement over conventional static dataset.*"

- *Use of proxy variables for hazard assessment:* The study estimates flood hazard using proxy variables derived from DEM, rainfall, and geomorphological data, rather than employing actual flood hazard data. More robust approaches, such as using satellite-based flood observations (e.g., Sentinel data) or hydrodynamic flood models, would provide a more accurate representation of flood hazard. Recent studies have successfully integrated observed flood data into multi-criteria decision-making (MCDM) models.

We acknowledge that actual flood observations—especially those derived from remote sensing or simulation—can offer valuable insights into the spatial extent of flooding event. While satellite datasets were not used as primary inputs for hazard modelling in our study, however, we did use Sentinel imagery and other remote sensing products for visual inspection and cross-referencing, especially in

areas with a known history of inundation. This was done to qualitatively validate the areas identified as high hazard zones through potential -based analysis. However, the choice of proxy variables in our study was based on specific considerations, as outlined below:

1. Limitations of Satellite-based flood observation: Sentinel-1 SAR data, while effective for flood detection due to its cloud-penetrating capabilities, has limited temporal resolution (typically 12-day revisit period over non-European countries). This makes it challenging to capture short-duration flash flood events unless coinciding with the satellite overpass. Optical datasets (e.g., MODIS, Landsat) provide higher temporal coverage but suffer significantly from cloud cover, especially during monsoon events when flood mapping is most needed. Moreover, mapping potential (rather than past or observed) flood hazard using satellite data would require long-term archival image analysis, which becomes methodologically and computationally intensive—particularly for a large study area (approximately 860,000 km²). Integrating multiple satellite sources to address temporal and spatial gaps would further increase the complexity and resource demands of the study, which was not feasible within the scope of this research.

2. Limitation of Hydrodynamic flood models: Hydrodynamic models require a variety of input data, including precipitation records, discharge data, and flow depth measurements, which are typically collected via rain gauges, river gauge stations, and hydrological datasets. In India, these datasets are often not readily available at the required spatial and temporal resolution for flood hazard modelling. Many regions lack publicly accessible, consistent, or high-quality rainfall and discharge data. In addition, hydrological station coverage may be sparse. The is typically managed by governmental agencies or research organizations, and accessing this information often requires formal requests or approvals. This process can be time-consuming, particularly for large geographic regions, and might involve bureaucratic delays, making it impractical within the scope and time constraints of this study.

Given the above limitations, we adopted a widely accepted approach using proxy variables (e.g., elevation, slope, drainage, rainfall) that are: Freely and readily available, relevant to flood generation processes and frequently validated in peer-reviewed flood risk assessments (e.g., AHP-based studies). While these proxies may not offer the same level of precision as hydrodynamic models, they have been effectively used in numerous flood risk assessments globally (as mentioned by the reviewer in the first comment), and they provide a reasonable alternative, especially when observed flood data is not available. These proxies offer consistent and scalable inputs for estimating flood risk, particularly in data-scarce and rapidly urbanising regions, and allow integration within the GIS-MCDM framework without compromising spatial coverage.

- *Methodological limitations and justification:* The traditional AHP model relies heavily on expert judgment, which introduces uncertainty. Recent studies have addressed this limitation by incorporating hybrid deep learning models and fuzzy AHP approaches, allowing for the integration of binary flood hazard data (e.g., flood-prone vs. non-flood-prone zones) into MCDM frameworks. Furthermore, quantitative validation and uncertainty analysis are essential to ensure confidence in results. The manuscript lacks a clear justification for the chosen methodology and does not address these recent methodological advancements.

We understand that the reviewer has two concerns- chosen methodology and recent methodological advancements related to AHP, and validation/uncertainty check.

1. **AHP as the preferred methodology for flood risk mapping and the recent advancements** – We would like to highlight the reasoning made in the earlier comment that the usage of AHP was intentional as it's a well-established method for flood risk studies. We acknowledge the fact that the AHP model depends on the expert judgment, but the parameters are chosen based on a literature review and expert judgements. The AHP method permits carrying out the analysis through assessing, integrating, additionally ranking of the various conflicting factors at a certain degree of information.

To emphasis the chosen methodology, we have modified the text in the manuscript.

Now, Lines 107 to 113 reads as *"Hydrodynamic Models are a subset of numerical models simulate the temporal and spatial variation of water flow and are widely used for flood forecasting and inundation mapping (Horritt & Bates, 2002). It requires a variety of input data, including precipitation records, discharge data, and flow depth measurements, which are typically collected via rain gauges, river gauge stations, and hydrological datasets. Many regions lack publicly accessible, consistent, or high-quality rainfall and discharge data. In addition, hydrological station coverage may be sparse. The data gathering can be time-consuming, particularly for large geographic regions, making it impractical within the scope and time constraints of this study."*

Lines 118-126 *"Given the above limitations, we adopted a multi-criteria decision-making (MCDM) approach using the Analytic Hierarchy Process (AHP), which allows for the integration of geomorphological, hydrometeorological, and socio-environmental factors. AHP was developed by Saaty (1980) is one of the widely known approaches for flood risk mapping (Sinha et al, 2008; Chakraborty and Mukhopadhyay, 2019; Ghosh and Kar, 2018; Grozavu, 2017; Huang et al., 2011; Mishra and Sinha, 2020). This approach uses pairwise comparisons to assess the extent to which one factor within the model is more important than the other, thereby producing a weighting for each factor. While we do not claim that this method is universally superior to hydrodynamic modelling, it offers a practical and intuitive framework for flood risk assessment in data-limited contexts. Empirical approaches such as MCDM have been widely used in flood studies and are considered effective when supported by robust spatial datasets and expert judgment (Teng et al., 2017)."*

To further address the recent methodological advancements related to AHP, we have added further text to the application of AHP in integration to other methods from line 145 to line 150 – "Fuzzy AHP (*FAHP) is one of the prevailing and powerful techniques that has been used for decision-making. Fuzzy set theory is the basis for doing FAHP (Xu et al., 2023 and Aggarwal et al., 2023). Mudashiru (2022) shows that the AHP and the FAHP methods applied are sensitive to model input change. Despite these variations, the flood hazard maps generated with the same factors and model presented almost similar maps to the sensitized maps. Furthermore, hybrid AI-MCDM models are emerging that combine the interpretability of MCDA with the computational intelligence of AI, thereby enabling more robust, data-driven, and context-sensitive decision-making frameworks for water resource planning (Gacu et al., 2025)."*

Also, the ration behind choosing AHP and not fuzzy AHP was using a new dataset in already established approach, which has been mentioned in the manuscript that the AHP is the most widely used MCDM method for flood hazard modelling.

2. **Quantitative validation and uncertainty analyses** have already been performed in the study. Validation and comparison with the impact-based data using GDIS and EMDAT has been explained in the manuscript in the section 3.7 and the results are described in the section 4.7. We have addressed the need to compare our results with an impact-based dataset, which in this case is the EMDAT and GDIS.

   As stated in the manuscript from line 679 to 682 – "*In Figure 12 we observe that the ROC-AUC curve plotted between the true positive rate (TPR) and false positive rate (FPR) has an area under the curve (AUC) value of 0.69 (or 69%), which is satisfactory – very good range as per the classification Das, 2020; Lin et al., 2019; Mukhtar et al., 2024; Roy et al., 2021; Saha and Agrawal, 2020)*". We are not sure what exactly the reviewer means "Furthermore, quantitative validation and uncertainty analysis are essential to ensure confidence in results".

---

## Author Comment (AC2)

**Reply to the Revier 2 comments:**

We would like to mention that the line numbers now mentioned in the comments are the new line numbers after revising the text

The manuscript *"Increasing Flood Risk in the Indian Ganga Basin: A Perspective from the Night-time Lights"* tackles an important issue—the escalating flood risk in the Ganga Basin—by proposing a methodology that integrates multiple geospatial datasets with NASA's Black Marble night-time lights as a proxy for human exposure using an AHP approach. While the approach and dataset integration are acceptable in principle, several conceptual and methodological issues must be addressed before the paper is suitable for publication.

Thank you for the comments and feedback on the research. Below is the reasoning for each comment as listed by the reviewer.

**Major Comments**

1. **Modelling Framework and Terminology**

- **Clarification of Model Types:**
  The manuscript currently confuses distinct modelling approaches. The authors refer to "physical," "numerical," and "hydrodynamic" models in ways that are inconsistent with established definitions. For example, physical models should be recognized as scaled, laboratory-based representations (e.g., flume models), while numerical models involve solving equations computationally. Hydrodynamic models, in contrast, specifically address the full dynamic wave Saint Venant equations in 1 or 2 dimensions derived from the Navier–Stokes framework in 3 dimensions. I strongly recommend that the authors clearly distinguish these models and ensure that each description is both scientifically accurate and well-referenced.

  We thank the reviewer for this insightful comment and fully acknowledge the need for precise and consistent terminology when referring to different modelling approaches. In response, we have revised the manuscript to clearly distinguish between physical, numerical, and hydrodynamic models in accordance with established definitions in the literature from line 103-117 and we very much hope this makes it clearer for readers. We stress that our own results and findings are not affected by the choice of terminology.

  *"A general overview of different techniques for mapping flood-prone areas is broadly categorised into physical, numerical and empirical approaches (Liu et al., 2024; Mukhtar et al., 2024; Teng et al., 2017). Physical Models are scaled-down laboratory representations (e.g., flume or basin models) used to study flow dynamics under controlled conditions (Heller, 2011; Hughes, 1993). Numerical Models simulate fluid flow using discretised solutions to governing equations such as the Saint Venant or Navier–Stokes equations (Teng et al., 2017). Hydrodynamic Models are a subset of numerical models simulate the temporal and spatial variation of water flow and are widely used for flood forecasting and inundation mapping (Horritt & Bates, 2002). It requires a variety of input data, including precipitation records, discharge data, and flow depth measurements, which are typically collected via rain gauges, river gauge stations, and hydrological datasets. Many regions lack publicly accessible, consistent, or high-quality rainfall and discharge data. In addition, hydrological station coverage may be sparse. The data gathering can be time-consuming, particularly for large geographic regions, making it impractical within the scope and time constraints of this study. Empirical Models in contrast, rely on observed historical data to identify statistical or heuristic relationships rather than simulating physical processes*

*(Teng et al., 2017). These include Multi-Criteria Decision-Making (MCDM) methods, which combine expert judgment with weighted spatial factors (e.g., slope, land use, rainfall) to assess flood susceptibility, statistical models like logistic regression and frequency analysis, and machine learning models that detect complex patterns from large datasets (Mosavi et al., 2018; Rahmati et al., 2016).*

*".*

- **Terminology Consistency:**
The terms "flood risk," "flood susceptibility," and "flood impacts" appear to be used interchangeably without precise definitions. This lack of clarity undermines the overall conceptual framework. I urge the authors to define these terms explicitly at the beginning of the manuscript and maintain consistent usage throughout the text.

*We sincerely thank the reviewer for highlighting this important issue. We agree that consistent and precise use of terminology is critical for clarity and conceptual rigor and we agree with the reviewer that we can improve this further. In response to this comment, we have made the following revisions and clarifications:*

1. *In line with the IPCC (2014) framework, we have now explicitly defined "flood risk" in the manuscript (lines 63–65) as follows: "According to the IPCC (2014) report, flood risk is defined as the potential for adverse consequences. It is a product of hazard, exposure and vulnerability based on the common framework adopted by the United Nations."*
2. *The term "susceptibility" was originally used in the manuscript in accordance with its general meaning: "the state or fact of being likely or liable to be influenced or harmed by a particular thing" (Oxford Dictionary). However, we acknowledge that in the context of flood risk assessments, susceptibility is more appropriately associated with hazard rather than risk. To prevent confusion, we have revised the manuscript to remove or rephrase the term where necessary. For example, the phrase: "Higher FHI means higher susceptibility to floods" has been revised to "Higher FHI means higher probability of flooding."*
3. *Following the IPCC, "impacts" refer to the consequences of realized risks on natural and human systems, including effects on lives, livelihoods, infrastructure, ecosystems, and more. In our study, the term "impacts" is specifically used in reference to data derived from EMDAT and GDIS, which record the actual consequences of flood events. If the reviewer's concern relates to the comparison made in Section 3.7, where modelled risk areas are validated against historical flood impact data from EMDAT, we would like to clarify that the terms are not used interchangeably in that context. Rather, we aim to assess whether areas identified as having high flood risk (i.e., areas with high hazard, exposure, and vulnerability) correspond to areas that have experienced actual flood impacts. We believe this validation step is appropriate and conceptually consistent with the definitions provided.*

   *We have revised the manuscript to ensure clear, consistent, and IPCC-aligned terminology throughout. We greatly appreciate the reviewer's comment, which helped improve the clarity and conceptual robustness of our work.*

- **Scientific Accuracy and Data Interpretation**
- **Rainfall Versus Precipitation:**
The manuscript contains statements suggesting that "rainfall" and "precipitation" are synonymous. However, precipitation is a broader term that includes snow, ice, and hail, while rainfall does not. This distinction is critical in the context of flood risk assessment, particularly in regions where non-rainfall

events may contribute to flooding. The authors should either revise this for accuracy or provide a strong justification for their interchangeable usage.

We thank the reviewer for highlighting this important distinction. In our study area, flooding is predominantly caused by **rainfall events**, and other forms of precipitation such as snow or hail do not significantly contribute to flood hazards. Our intention was to refer specifically to rainfall throughout the manuscript.

To maintain terminological accuracy and consistency, we have revised the manuscript to replace instances of "precipitation" with "rainfall" where appropriate. We believe this revision improves clarity and aligns the language of the manuscript more closely with the physical processes relevant to the study area.

- **Justification of Dataset Choices:**
  The choice of datasets, particularly the ASTER-GDEM and CHIRPS data, is a subject of concern. Several studies indicate that higher-resolution DEMs (e.g., those derived from Copernicus DEM) might offer more reliable vertical accuracy.
- Similarly, while CHIRPS is a robust dataset, some recent literature suggests that GPM-IMERG may provide superior performance in monsoon-dominated regions. The manuscript would benefit from a more detailed discussion on the selection of these datasets and any limitations they might introduce.
- I am also really confused by the choice of the annual nighttime lights dataset as a proxy, even when the NASA Black Marble product includes daily products (https://doi.org/10.1016/j.rse.2018.03.017) which would correspond much better to flood recovery processes (see here: https://doi.org/10.1016/j.rse.2025.114645). Would the yearly averaged product not smooth out the impacts in most areas, defeating the purpose or the proxy? Or are the authors claiming only to examine floods where no recovery was possible within a year to be able to see this impact on an annual scale? In any case, this is the main novelty of the paper apparently so I strongly recommend better defending this methodological choice and how this may influence their conclusions.

Thank you for raising these important points regarding the selection of datasets and the use of NASA's Black Marble Nighttime Lights (NTL) product.

**1. On the choice of ASTER-GDEM and CHIRPS data:** We acknowledge that higher-resolution DEMs such as the Copernicus DEM offer improved vertical accuracy and could enhance hazard modelling, particularly for small-scale hydrological analyses. However, in our study, we selected ASTER-GDEM at 1 arc-second (~30 m) resolution due to its compatibility with the spatial resolution of other datasets—especially the NTL dataset, which has a coarser resolution (~500 m, or 15 arc-seconds). The objective was to harmonise all layers to a common spatial resolution for weighted overlay and minimise interpolation errors during resampling. Given that our analysis focuses on regional-scale patterns across the large expanse of the Ganga Basin, the vertical resolution of ASTER-GDEM was considered sufficient for capturing broad-scale hydro-geomorphic variations.

Similarly, CHIRPS was selected for precipitation data due to its long temporal coverage, high spatiotemporal resolution (0.05°), and robust performance in data-scarce regions. While we recognise the increasing use of GPM-IMERG, comparative studies show mixed results in monsoon-dominated areas, with some favouring CHIRPS for its bias-corrected historical performance and better agreement with rain gauge data over India. Nonetheless, we acknowledge this limitation and will include a brief comparative discussion in the revised manuscript to clarify our dataset choice and its implications.

**2. On the use of annual Nighttime Lights (NTL) data:** We appreciate the reviewer's insight and agree that the daily NTL product has great utility in disaster response and recovery assessments. However, our study focuses on long-term flood risk assessment, not short-term flood impact or recovery dynamics. The primary goal was to quantify average annual exposure to flood risk, akin to how demographic datasets (e.g., census-based population) are used in many existing studies. For this purpose, the annual NTL product provides a consistent, smoothed estimate of human presence and economic activity over time.

While we acknowledge that daily NTL could offer finer temporal sensitivity, it is also subject to limitations such as cloud cover, moonlight interference, and post-disaster power disruptions, making it less reliable for long-term trend analyses. The yearly NTL product, by averaging over these daily variations, offers a stable proxy for human settlement patterns, particularly useful when examining multi-annual trends in exposure across a large and diverse region like the Ganga Basin.

Moreover, a key novelty of our work lies in leveraging a decade-long time series of annual NTL data to track exposure evolution—a capability not offered by static population datasets. This supports a more dynamic and spatially resolved assessment of flood risk, aligning with our study's aim to map changing flood risk patterns over time, rather than evaluate isolated flood events. This point has been highlighted in the manuscript from line 83 to 89 – "*The key novelty of the approach lies in using night-time lights (NTL) as a proxy for flood exposure within the basin, unlike the population data, and leveraging the temporal availability of the data. NTL data is collected from satellite-based sensors like VIIRS (Visible Infrared Imaging Radiometer Suite) which offers a dynamic, consistent, and spatially explicit view of human activity and settlement patterns. Over the past decade, NTL data have been increasingly used to monitor urban expansion, economic activity, and disaster impacts (Andries et al., 2023; Román et al., 2018; Wang et al., 2018). The NTL data can reflect the real-time distribution of human activities at a large scale and with better temporal frequency, compared to traditional statistics and census data (Fang et al., 2021)*".

- **Methodological Presentation**
- **AHP Framework:**
  The use of the Analytical Hierarchy Process (AHP) for integrating multiple flood risk factors is not novel even though it is widely used and acceptable in this case. However, **the manuscript would benefit from clearer descriptions of how the pairwise comparisons were conducted, how consistency was ensured, and how the resulting weights were validated**. In some instances, the paper appears to overcomplicate the presentation of these steps. A more concise explanation, supported by relevant references, would improve readability.

  Thank you for your observation regarding the use of the Analytical Hierarchy Process (AHP) and its methodological clarity.

  We would like to respectfully point out that the manuscript already provides a clear and structured explanation of the AHP implementation. Section 3.2 (Lines 272-275) outlines the four core stages of the AHP process: (i) parameter hierarchy construction, (ii) pairwise comparison matrix, (iii) weight normalisation, and (iv) consistency check, supported by appropriate literature (Ghosh and Kar, 2018; Mishra and Sinha, 2020; Roy et al., 2021). Unlike many studies that treat these steps in a generalised manner, we have deliberately broken down and described each phase in detail to enhance transparency and reproducibility.

*After discussion with the co-authors, we believe the current level of detail is appropriate and sufficient for understanding the methodology. However, we have edited the section lightly to make sure things are as readable as possible.*

- **Exposure and Vulnerability**
- While the use of night-time lights as a proxy for human exposure is interesting, the manuscript does not sufficiently differentiate exposure from vulnerability. In several sections, vulnerability data (e.g., district-level vulnerability indices) are used in contexts that suggest they represent exposure. A thorough re-examination and clear delineation of these concepts are needed.

  *The definition of exposure and vulnerability is already described in the manuscript from line 69 to 74.*
  - *"Exposure is defined as the presence of people, livelihoods, species or ecosystems, environmental functions, services, and resources, infrastructure, or economic, social, or cultural assets in places and settings that could be adversely affected",*
  - *"Vulnerability is defined as the propensity or predisposition to be adversely affected. Vulnerability encompasses a variety of concepts and elements including sensitivity or susceptibility to harm and lack of capacity to cope and adapt".*

  *The important aspect of this research was to identify hazard, exposure and vulnerability and differentiate between each of them in influencing the flood risk. We are not sure how the reviewer thinks that the vulnerability and exposure have been used interchangeably.*

  *However to highlight the indicators of vulnerability index used in the Climate Vulnerability Assessment Report produced by the Department of Science and Technology, Government of India, we have added the names of the indicators used in section 3.1 from line 242 to 248 – "The indicators used in the report were - Percentage of population living below the poverty line (BPL), income share from natural resources, share of horticulture in agriculture, proportion of marginal and small landholdings, women's participation in the workforce, Yield variability of food grains, area under rainfed agriculture, forest area per 1000 rural population, incidences of vector- borne diseases and water-borne diseases, Area covered under centrally funded crop insurance schemes (such as Pradhan Mantri Fasal Bima Yojna (PMFBY) and Revised Weather-based Crop Insurance, Scheme (RWBCIS), implementation of Mahatma, Gandhi National Rural Employment Guarantee, Act (MGNREGA), road and rail-network, the density of healthcare workers".*

**Minor Comments**

- **Figure and Visual Clarity:**
  Some figures, such as the workflow diagram and flood hazard maps, are not sufficiently legible at 100% zoom. I recommend that the authors provide higher-resolution images or ensure that all text and symbols are easily readable.
  *Thank you for the feedback. The figures have been revised.*

- **Scale and Resolution Issues:**
  There are questions regarding the spatial and temporal resolutions used, the rainfall data are daily, aggregated to monthly scales – the aggregation method is not specified – and then kept just for the monsoon months but then is compared to annual scale night-time lights data, which does not really make sense from my point of view.

The rainfall data used in the study were originally available at a daily temporal resolution. These were aggregated to monthly totals using a summation method and then averaged over the five-month monsoon period (June to October) for each year. This period was selected because it corresponds to the primary flood season in the study area, during which most flood events occur. We have now explicitly stated the aggregation method and rationale in the revised manuscript.

However, to be precise in the manuscript, we have revised the text and added the aggregation method from line 331 to 334 – *"For rainfall analysis, we used the daily data CHIRPS raster data to prepare the monthly total rainfall. These were aggregated to monthly totals using a summation method and then averaged over the five-month monsoon period (June to October) for each year. This period was selected because it corresponds to the primary flood season in the study area, during which most flood events occur."*

Regarding the use of annual night-time lights (NTL) data, we acknowledge the temporal mismatch with the monsoon-season rainfall data. However, the NTL data in this study were not used to capture short-term flood dynamics but rather to serve as a proxy for human settlement patterns and population distribution, which are relatively stable over the course of a year. This approach is consistent with previous studies that have used NTL data as a spatial indicator of exposure or vulnerability in flood risk assessments (e.g., Ceola et al., 2014)

- **References:**
  Several assertions in the manuscript lack adequate citation. For example, claims about the effectiveness of different modelling approaches and the influence of rainfall intensity on flooding should be supported by additional literature.
  Thank you for the feedback. This has been mentioned in the previous comments, and we have tried our best now to address the different modelling approaches and influence of rainfall on flooding.

In its current form, the manuscript has several conceptual ambiguities and methodological inconsistencies that must be resolved before publication. Addressing these concerns will substantially strengthen the manuscript and enhance its contribution to the field of flood risk assessment.

---

## Author Comment (AC3)

**Reply to Reviewer 3 comments:**

We would like to mention that the line numbers now mentioned in the comments are the new line numbers after revising the text.

**Title: Increasing flood risk in the Indian Ganga Basin: A perspective from the night-time lights**

This study assessed the flood hazard, exposure, vulnerability and risk level of the Indian Ganga Basin based on a multi-criteria risk assessment methodology (hierarchical analysis) using selected hazard, exposure and vulnerability assessment indicators. The novelty of the study lies in using night-time lights as a proxy for exposure within the basin, unlike the conventional population data.

I have a few comments/questions:

- The abstract section of the manuscript is illogical and lengthy. For example, the innovation of the paper is highlighted twice in the beginning and the end of the abstract. It is recommended that the authors rewrite the abstract section according to research background, methods and content, results, validation, innovativeness and application value.

  Thank you for the feedback. We have rewritten the abstract as suggested. The new abstract is –

  *"Floods in the Ganga Basin, one of India's most densely populated and geomorphologically dynamic regions, have intensified in frequency and severity due to changing climatic regimes, extreme rainfall events, and rapid land-use transformations. Understanding the spatial and temporal evolution of flood risk—arising from the interaction between hazard, exposure, and vulnerability—is critical for evidence-based risk management.*

  *Here, we present a spatially explicit flood risk assessment using the Analytical Hierarchy Process (AHP), which is a Multi-Criteria Decision Making (MCDM) model. The methodology integrates remote sensing and geospatial datasets to map flood risk using flood hazard and vulnerability, while introducing a novel exposure metric based on NASA's Black Marble Nighttime Lights (NTL) data. Unlike traditional demographic proxies, the NTL dataset captures temporally resolved patterns of human presence and economic activity, enabling dynamic representation of exposure at large spatial scales.*

  *Our results indicate a marked escalation in flood risk across the eastern Ganga Basin, with high-risk zones concentrated in Bihar, eastern Uttar Pradesh, eastern Madhya Pradesh, and northern West Bengal. Validation against historical flood impact data from the EM-DAT and GDIS databases yields an accuracy of ~70%, underscoring the model's robustness. The analysis reveals increasing human exposure and shifts in rainfall intensity as dominant drivers of emerging flood risk hotspots. This study leverages the temporal availability of the data, enabling a real-time distribution of human activities at a large scale and with greater temporal resolution.*

  *This study demonstrates the applicability of dynamic, satellite-derived exposure indicators within an established flood risk assessment framework. The approach facilitates scalable, data-driven risk mapping with relevance for anticipatory planning and regional-scale disaster risk reduction across transboundary flood-prone landscapes".*

- The confusing use of professional terms such as "flood risk" and "flood susceptibility" means that the professional level of research manuscripts needs to be improved.

  We have carefully clarified the use of the terms flood risk and flood susceptibility, as we have explained above in response to Reviewer 2

- I disagree with the discussion of the superiority of the methodology of this study in lines 652-655 of the manuscript. Accurate flood risk assessment results are obtained by driving a hydrological-hydraulic model to capture flood hazard, and then combining exposure, vulnerability, and the level of prevention and mitigation. The use of multi-criteria methods to assess flood risk levels in this study may be limited by data and technical expertise. Hydrological and hydrodynamic models can simulate the depth and range of flooding and obtain more accurate risk assessment results.

  Lines 652-655: "Our flood risk map of the Ganges basin was developed using an integrated hydrogeological approach and AHP methodology, which is superior to traditional hydrological and hydraulic modelling as it combines physical (geomorphological) criteria with hydrometeorological data. This emphasises a process-based understanding that overcomes the need for intensive hydrological data required for flood hydraulic models (Mishra and Sinha, 2020)".

  We thank the reviewer for this important observation and fully acknowledge the strengths of hydrological and hydrodynamic models in simulating flood depth, extent, and hazard with high precision. We agree that such models are essential tools in flood risk assessment, particularly when high-resolution, long-term hydrological data are available.

  However, as noted in the revised manuscript from lines 107-113, *"Hydrodynamic Models are a subset of numerical models that simulate the temporal and spatial variation of water flow and are widely used for flood forecasting and inundation mapping (Horritt & Bates, 2002). It requires a variety of input data, including precipitation records, discharge data, and flow depth measurements, which are typically collected via rain gauges, river gauge stations, and hydrological datasets. Many regions lack publicly accessible, consistent, or high-quality rainfall and discharge data. In addition, hydrological station coverage may be sparse. The data gathering can be time-consuming, particularly for large geographic regions, making it impractical within the scope and time constraints of this study."*

  *Line 118-126: "Given the above limitations, we adopted a multi-criteria decision-making (MCDM) approach using the Analytic Hierarchy Process (AHP), which allows for the integration of geomorphological, hydrometeorological, and socio-environmental factors. AHP was developed by Saaty (1980) and is one of the widely known approaches for flood risk mapping (Sinha et al, 2008; Chakraborty and Mukhopadhyay, 2019; Ghosh and Kar, 2018; Grozavu, 2017; Huang et al., 2011; Mishra and Sinha, 2020). This approach uses pairwise comparisons to assess the extent to which one factor within the model is more important than the other, thereby producing a weighting for each factor. While we do not claim that this method is universally superior to hydrodynamic modelling, it offers a practical and intuitive framework for flood risk assessment in data-limited contexts. Empirical approaches such as MCDM have been widely used in flood studies and are considered effective when supported by robust spatial datasets and expert judgment (Teng et al., 2017).*

  Accordingly, we have revised the original statement in lines 652–655 (original manuscript) to reflect a more balanced and accurate comparison in line 700-703 - *"Our flood risk map of the Ganges basin was developed using an integrated hydrogeological approach and AHP methodology, which offers a practical alternative to traditional hydrological and hydraulic modelling in data-scarce regions. By*

*combining physical (geomorphological) criteria with hydrometeorological data, this approach supports a process-based understanding of flood risk where intensive hydrological datasets are unavailable (Mishra and Sinha, 2020).*

We hope this clarification addresses the reviewer's concern and improves the scientific accuracy and objectivity of the manuscript.

- The vulnerability assessment in the study used existing vulnerability products, the reliability of which was not verified. In addition, which indicators considered in the vulnerability assessment were not clearly given.

We thank the reviewer for raising this important point regarding the vulnerability assessment. In our study, the vulnerability component of flood risk was derived from the **Climate Vulnerability Assessment Report- DST 2020**, developed by the **Department of Science and Technology (DST), Government of India**. We have now clarified in the manuscript the indicators used for developing the vulnerability index from lines 240-246. However, the methodological details to evaluate whether the Indian government's vulnerability assessment is or is not fully reliable are beyond the scope of this paper.

- The novelty of the work lies in using night-time lights as a proxy for exposure within the basin, unlike the conventional population data. However, in the flood risk assessment framework based on multi-criteria methods, using night light data instead of population and economic data as an innovation in flood risk assessment seems to be somewhat insufficient for the entire study, but it is acceptable.

In flood risk assessment, research innovation would be more prominent if hazard assessment used flood inundation information obtained based on remote sensing data, while taking into account changes in flood vulnerability.

We thank the reviewer for this constructive feedback and for acknowledging the novelty of using night-time lights (NTL) data as a proxy for exposure in flood risk assessment. We agree that integrating flood inundation data derived from remote sensing or hydrodynamic models could further enhance the robustness of hazard assessment. This is indeed a valuable direction for future research, which was also suggested by Reviewer 1.

However, Sentinel-1 SAR data, while effective for flood detection due to its cloud-penetrating capabilities, has limited temporal resolution (typically 12-day revisit period over non-European countries). This makes it challenging to capture short-duration flash flood events unless they coincide with the satellite overpass. Optical datasets (e.g., MODIS, Landsat) provide higher temporal coverage but suffer significantly from cloud cover, especially during monsoon events when flood mapping is most needed. Moreover, mapping potential (rather than past or observed) flood hazard using satellite data would require long-term archival image analysis, which becomes methodologically and computationally intensive, particularly for a large study area (approximately 860,000 km²). Integrating multiple satellite sources to address temporal and spatial gaps would further increase the complexity and resource demands of the study, which was not feasible within the scope of this research.

Regarding the dynamic nature of vulnerability, we fully agree that incorporating temporal changes in vulnerability would significantly enrich flood risk assessments. However, such an approach would require high-resolution, time-series socio-economic and infrastructural data, which are currently limited

in availability and consistency across the study region. We acknowledge this as a promising area for future research and have added a note to this effect in the revised manuscript.

The dynamic trends of flood risk levels should be placed in the results section of the study rather than the discussion section, which should focus on the superiority of using nighttime light data for flood exposure and risk assessment.

We sincerely thank the reviewer for this thoughtful suggestion. We carefully considered the placement of the dynamic trends of flood risk levels during the manuscript development process, including extensive discussions among co-authors. Our rationale for presenting these trends in the Discussion section is that they are not standalone empirical results derived directly from the modelling outputs, but rather an interpretive synthesis of observed spatial and temporal patterns about socio-environmental dynamics. We intended this section to serve as a bridge between the quantitative results and their broader implications for flood risk management, policy, and exposure assessment.

We explored putting this in the results, but on balance, given this interpretive nature, we believe the **Discussion** section is the most appropriate place for this content. However, we appreciate the reviewer's perspective.